# p300/CBP is an essential driver of pathogenic enhancer activity and gene expression in Ewing sarcoma

Laura C Godfrey [ID][1], Brandon Regalado[1], Sydney R Schweber [ID][1], Charles Hatton [ID][1], Daniela V Wenge [ID][1], Yanhe Wen [ID][1], Meaghan Boileau[1], Maria Wessels[1,2], Jun Qi[3], Christopher J Ott[4,5,6], Kimberly Stegmaier[1,5], Miguel N Rivera[5,7] & Scott A Armstrong [ID][1✉]

## Abstract

The t(11;22) translocation encodes the EWS::FLI1 fusion oncoprotein which is the primary driver of Ewing sarcoma. EWS::FLI1 creates unique, de novo pathogenic enhancers that drive gene expression and are a central mechanism of oncogenesis. Which chromatin regulatory proteins are critical to this mechanism is understudied. Here, we perform a comparative analysis of the function of the chromatin complexes MLL3/4 and p300/CBP in EWS::FLI1-mediated gene regulation. Using EWS::FLI1 degradation models, we define a subset of EWS::FLI1-sensitive enhancers whose activity correlates with p300/CBP function. We perturb both chromatin complexes to establish that in contrast to MLL3/4, p300/CBP is a critical regulator of EWS::FLI1-driven enhancer activity and downstream gene expression. We also show that p300/CBP small-molecule inhibition decelerates tumor growth in vivo. Our work highlights the context-dependent nature of chromatin protein activity at oncogenic enhancers and reveals p300/CBP as an important regulator of Ewing sarcoma.

**Keywords** EWS::FLI1; Enhancers; Histone Modifications; Ewing Sarcoma; p300/CBP

**Subject Categories** Cancer; Chromatin, Transcription & Genomics

## Introduction

Many cancers driven by oncogenic fusion proteins involve epigenetic alterations, where transcriptional machinery can be hijacked by the fusion protein to create a pathogenic gene expression landscape. The molecular mechanisms by which oncogenic fusion proteins modulate gene expression through chromatin-associated complexes remain largely unexplored, yet this area of research holds significant promise for therapeutic advancements. This potential has been most prominently explored in the context of Mixed Lineage Leukemia (MLL)-rearranged leukemia, where mechanistic insight has led to the development of small-molecule inhibitors targeting the Menin-MLL interaction which are showing success in the clinic (Krivtsov et al, 2019). Despite this progress, there is a lack of understanding in other fusion-driven cancers regarding the identification, and mechanisms, of critical chromatin complexes involved in disease pathology.

A notable example of this is Ewing Sarcoma, a highly aggressive form of pediatric cancer, arising in the bone and soft tissue, which has no current targetable therapies (Grünewald et al, 2018). In over 85% of cases, the resulting fusion protein consists of the intrinsically disordered $NH_3$-terminal domain of Ewing sarcoma breakpoint region 1 protein (EWSR1) and the COOH-terminal DNA-binding domain of Friend leukemia virus integration 1 protein (FLI1), an Erythroblast Transformation Specific (ETS) transcription factor, creating EWS::FLI1.

To promote disease formation, EWS::FLI1 exhibits a unique ability permitting it to bind in tandem to otherwise non-functional, non-conserved GGAA repetitive motifs throughout the genome (Johnson et al, 2017; Gangwal et al, 2008). EWS::FLI1 is sufficient to induce chromatin accessibility at GGAA repeat regions consistent with its function as an oncogenic pioneer transcription factor (Boulay et al, 2018; Riggi et al, 2014, 2008). Upon binding, EWS::FLI1 initiates the activation of pathogenic de novo enhancers which function as cis-regulatory regions in the genome that support active transcription of their associated gene. Activation of EWS::FLI1 enhancers leads to upregulation of nearby genes which are highly expressed in Ewing sarcoma tumors and critical for growth (Smith et al, 2006; Sanchez et al, 2008; Riggi et al, 2014).

Because EWS::FLI1 enhancers are a unique set of pathogenic, non-conserved cis-regulatory regions their regulatory mechanisms are unknown. While this subset of enhancers has been shown to share features resembling wild-type, active enhancers including

[1]Department of Pediatric Oncology, Dana-Farber Cancer Institute, and Division of Hematology/Oncology, Boston Children's Hospital, and Harvard Medical School, Boston, MA, USA. [2]Department of Hematology, Hemostasis, Oncology and Stem Cell Transplantation, Hannover Medical School (MHH), Hannover, Germany. [3]Department of Cancer Biology, Dana-Farber Cancer Institute, Boston, MA, USA. [4]Massachusetts General Hospital Cancer Center, Charlestown, MA, USA. [5]Broad Institute of MIT & Harvard, Cambridge, MA, USA. [6]Harvard Medical School, Boston, MA, USA. [7]Department of Pathology, Massachusetts General Hospital, Boston, MA, USA. ✉E-mail: Scott_Armstrong@dfci.harvard.edu

histone 3 lysine 4 monomethylation (H3K4me1) and histone 3 lysine 27 acetylation (H3K27ac), the importance of these modifications for enhancer activity is unstudied (Riggi et al, 2014; Tomazou et al, 2015). The deposition of these histone modifications are facilitated by Lysine (K)-specific methyltransferases KMT2C and KMT2D (referred to throughout this study as Mixed Lineage Leukemia 3 and Mixed Lineage Leukemia 4, MLL3/4) and the E1A binding protein (p300) and CREB binding protein (CBP) (Bannister and Kouzarides, 1996; Ogryzko et al, 1996). The MLL3/4 complex has been shown to play a role in both enhancer maintenance and enhancer poising in different cellular contexts (Dorighi et al, 2017; Lee et al, 2013; Wang et al, 2016; Yan et al, 2018; Wang et al, 2017). Similarly, both p300 and CBP proteins serve as important co-activators in wild-type enhancer regulation and transcriptional activity (Durbin et al, 2022; Narita et al, 2021; Sen et al, 2019). While context-dependent functional differences between MLL3 and MLL4, as well as between p300 and CBP, have been highlighted and may exist in the Ewing sarcoma context, we will refer to these proteins collectively as MLL3/4 and p300/CBP throughout the paper, given the focus of dual targeting of these proteins in our study.

Both p300 and the MLL3/4 complex component, WDR5, have been shown to co-occupy EWS::FLI1 enhancer regions (Riggi et al, 2014). Moreover, EWS::FLI1 is sufficient to recruit p300 and WDR5 when overexpressed in mesenchymal stem cells (Riggi et al, 2014). Despite this evident reliance on EWS::FLI1 for the recruitment of p300 and MLL3/4 complex to EWS::FLI1 enhancers, the contribution of these chromatin modifiers to active enhancer maintenance and downstream gene expression remains unclear. In this study, we conducted a comprehensive assessment of the roles played by H3K4me1 and H3K27ac modifications, along with their associated chromatin complexes, p300/CBP and MLL3/4, in regulating EWS::FLI1 enhancer activity and the subsequent gene regulatory network. By dissecting the function of these complexes at these unique regulatory regions, we aimed to gain a deeper mechanistic insight into how the EWS::FLI1 fusion protein orchestrates the aberrant gene expression pattern that drive the formation of this disease and provide potential opportunities for therapeutic development.

## Results

### p300 and CBP occupy EWS::FLI1 enhancer regions and correlates with EWS::FLI1 binding intensity

To initially study the role of p300/CBP and MLL3/4 complex, we aimed to generate a complete assessment of complex occupancy at EWS::FLI1 enhancers. 1785 EWS::FLI1 enhancer regions have been characterized in previous studies in Ewing sarcoma cell lines (Riggi et al, 2014; Sanalkumar et al, 2023). FLI1 Chromatin Immunoprecipitation-sequencing (ChIP-seq) verified the presence of EWS::FLI1 at these pre-defined regions in this set of experiments (Fig. EV1A). In addition, motif analysis of FLI1 ChIP-seq confirmed an enrichment at FLI1 transcription factor motifs (Fig. EV1B). Next, we assessed p300/CBP and MLL3/4 complex binding in the Ewing sarcoma cell line A673 using ChIP-seq. This analysis encompassed p300 and CBP as well as all members of the MLL3/4 complex for which there was a commercially available

ChIP-grade antibody, including MLL4, WDR5, RbBP5, ASH2L, PTIP1 and UTX. We observed that all factors were bound throughout the genome includive of intergenic and gene body regions indicative of enhancer-associated roles (Fig. EV1C). Consistent with previous findings, we confirmed that EWS::FLI1 enhancers were occupied by p300 and H3K27ac (Riggi et al, 2014) (Figs. 1A and EV1E). This was accompanied by concurrent CBP binding revealing that both acetyltransferases are likely active at these pathogenic enhancers (Fig. 1A). Similarly, we observed occupancy of all MLL3/4 complex members tested (Fig. 1B). This included MLL4 itself as well as common (WDR5, RbBP5, ASH2L) and unique (UTX and PTIP1) complex components and this was coupled with both H3K4me1 and H3K4me2 (Figs. 1B and EV1E). Binding of these factors was exemplified at well-characterized enhancer loci, including NKX2-2 and CCND1 (Fig. 1C). Furthermore, motif analysis of p300, CBP and MLL4 ChIP highlighted an enrichment at FLI1 and other ETS transcription factor motifs (Fig. EV1D). Of note, we observed H3K4me3 at only a small number of EWS::FLI1 enhancers, consistent with H3K4me3 being a modification predominantly associated with promoter activity (Fig. EV1E,F).

To assess chromatin protein occupancy in a quantitative manner, we compared the number of MACS2-defined ChIP-seq peaks for each factor at the EWS::FLI1 enhancer regions. We observed that p300 and CBP called peaks overlapped with 86% (1450/1785) and 94% (1590/1785) of EWS::FLI1 enhancers, respectively indicating that both acetyltransferases are likely active at EWS::FLI1 enhancer regions (Fig. 1D). Interestingly, p300 and CBP were both highly expressed in all Ewing sarcoma cell lines assessed in this study further supporting a role for both proteins at EWS::FLI1 enhancers (Fig. EV1G).

In comparison, robust binding of unique components of the MLL3/4 complex, including MLL4 and UTX, were overall detected at fewer EWS::FLI1 enhancer regions, 25% and 16%, respectively (Fig. 1D). Of note, most enhancer regions were predominantly modified with both H3K4me1 and H3K4me2, suggesting that MLL3 may be playing a compensatory role at non-MLL4 bound EWS::FLI1 enhancers. Next, we examined the correlation of ChIP-seq peak intensities at EWS::FLI1 enhancer regions of EWS::FLI1 itself with other factors. We observed a robust positive correlation ($R^2 = 0.66$) between binding of p300 and EWS::FLI1 (Fig. 1E). This contrasts with MLL4, UTX and WDR5 which demonstrated a weaker correlation ($R^2 = 0.06$, 0.12 and 0.04, respectively) with EWS::FLI1 binding levels (Fig. 1E). Taken together, these findings suggest that p300/CBP may be more closely linked to EWS::FLI1 occupancy, indicating that p300/CBP potentially play a more central role in EWS::FLI1-dependent enhancer regulation compared to the MLL3/4 complex.

### EWS::FLI1 degradation reveals a subset of sensitive EWS::FLI1 gene targets and enhancers

Given our hypothesis that both p300 and CBP play a central role in EWS::FLI1 enhancer regulation, we aimed to closely examine the extent to which p300/CBP, compared to MLL3/4, are dependent on and linked to EWS::FLI1 occupancy and transcriptional activity at enhancers. PROTAC technology has recently emerged as a powerful tool in other cancer models, enabling the investigation of the immediate consequences of acute onco-fusion protein loss on

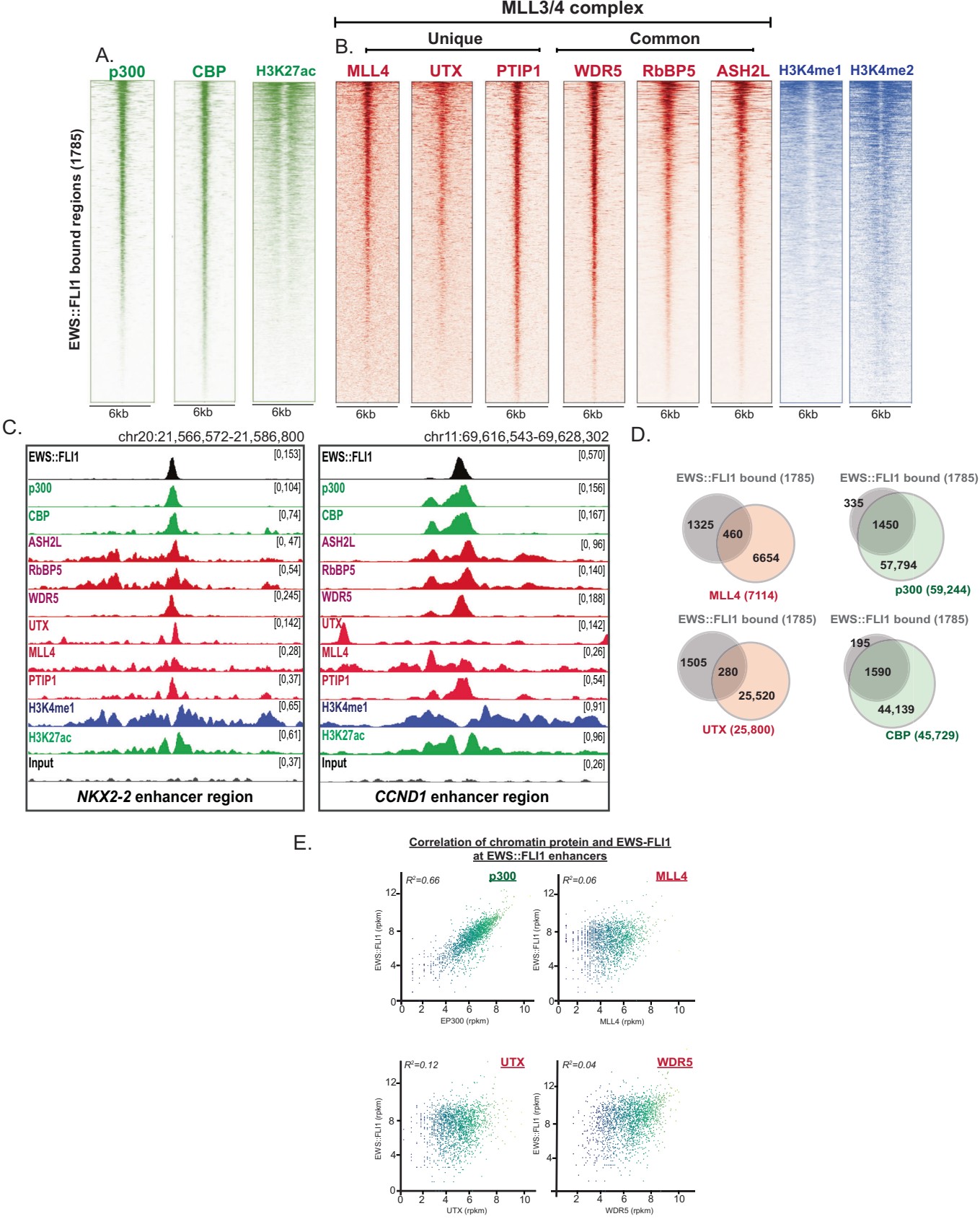

◄ **Figure 1. p300/CBP occupy EWS::FLI1 enhancer regions and correlate with EWS::FLI1 binding intensity.**

(A) Tornado plots of p300, CBP and H3K27ac ChIP-seq in A673 cells at EWS::FLI1 enhancer regions. (B) Tornado of MLL3/4 complex members and H3K4me1/2 ChIP-seq in A673 cells at EWS::FLI1 enhancer regions. (C) ChIP-seq tracks at *NKX2-2* enhancer and *CCND1* enhancer. Enhancer regions highlighted in orange. (D) MACS2 peak overlaps of p300, CBP, MLL4, and UTX with EWS::FLI1 enhancer regions. (E) Correlation of peak intensity of EWS::FLI1 with p300, MLL4, UTX, and WDR5. R2 value for each correlation shown on each graph.

chromatin dynamics and gene expression (Olsen et al, 2022; Zhang et al, 2022). To evaluate the immediate and direct effects of disrupting EWS::FLI1 chromatin binding, we employed two complementary PROTAC degradation systems: the EWS::FLI1-dHalo cell line (A673-dHALO) and the FKBP-EWS::FLI1 cell line (EWS-502 dTAG) (Chong et al, 2022; Nabet et al, 2020) We further compared these approaches with CRISPR-Cas9-mediated genetic deletion of EWS::FLI1, anticipating that the latter might reveal longer-term compensatory or indirect mechanisms of action.

We confirmed reduced EWS::FLI1 protein levels over a 24-hour time course in both degrader cell lines, and following seven days of genetic deletion with two independent gRNAs (Fig. EV2A). This decrease was matched by loci-specific perturbation in EWS::FLI1 binding using ChIP-qPCR at the *NKX2-2* enhancer as rapidly as 1–3 h post degradation (Fig. EV2B,C). To initially understand the dynamics between EWS::FLI1 binding itself and transcriptional changes, we performed RT-qPCR to measure enhancer RNA (eRNA), indicative of enhancer activity (Kim et al, 2010), for *NKX2-2* eRNA and mRNA from the downstream gene. We observed a time-dependent decrease in transcription of *NKX2-2* eRNA and mRNA after three hours of EWS::FLI1 degradation in both degrader systems (Fig. EV2D,E). This indicates that reduced EWS::FLI1 binding is almost immediately coupled with reduced *NKX2-2* enhancer activity and downstream gene expression. This loss of transcription was more severe following EWS::FLI1 genetic deletion, confirming that prolonged abrogation of EWS::FLI1 binding may induce a cascade of events that leads to the eventual collapse of transcription to almost undetectable levels (Fig. EV2F).

To evaluate the rapid changes in gene expression in a genome-wide manner following EWS::FLI1 degradation, we performed RNA-sequencing (RNA-seq) analysis at 24 h post degradation in both degrader systems. Gene Set Enrichment Analysis (GSEA) confirmed a significant downregulation of EWS::FLI1-associated gene expression (Fig. EV2G). We assessed all downregulated genes following EWS::FLI1 degradation and identified 3157 downregulated genes in the A673-dHALO system and 2026 downregulated genes in the EWS-502 dTAG system (Fig. 2A). Among these downregulated genes, we found 159 and 198 genes associated with EWS::FLI1 enhancers in the A673-dHALO and EWS-502 systems, respectively (Fig. 2A; Tables EV1 and 2). Going forward, we refer to these genes and their associated enhancer(s) as 'EWS::FLI1 sensitive'. Importantly, 62 downregulated genes were observed in both degrader systems and included well-characterized EWS::FLI1 targets such as *NKX2-2* and *VRK1* (Fig. EV2H). Using ribo-depleted RNA-seq, we were able to detect eRNA produced from EWS::FLI1 enhancer regions. At EWS::FLI1-sensitive enhancers, we observed a significant reduction in eRNA levels compared to all other unperturbed EWS::FLI1-regulated enhancers after 24 h of degradation (Fig. 2B; Dataset EV1). Furthermore, evidence of this reduction was observed as early as three and six hours after degradation at example EWS::FLI1-regulated enhancers in the

A673-dHALO system (Fig. 2C,D; tracks). Taken together, these results demonstrate that the EWS::FLI1 binding is dynamically linked to transcriptional maintenance from both the enhancer and gene target of a subset of enhancers and genes highly sensitive to loss of EWS::FLI1 occupancy.

## Rapid degradation of EWS::FLI1, and loss of enhancer transcription, is acutely linked to reduced p300/CBP occupancy and H3K27ac levels but not H3K4me1

As maintained transcription is tightly coupled to EWS::FLI1 binding at sensitive enhancers, we next wanted to understand how linked p300/CBP and MLL3/4 activity and occupancy were with these events. First, we assessed H3K27ac and H3K4me1 levels at the *NKX2-2* enhancer following EWS::FLI1 degradation. We observed a rapid reduction of H3K27ac after only 3–6 h of degradation, closely linked with loss of EWS::FLI1 occupancy and decreased eRNA levels (Figs. 2E and EV2I). In contrast, we observed no perturbation in H3K4me1 levels, even at 24 h post degradation where gene expression was reduced by approximately 80% by RT-qPCR (Fig. 2E). This suggests that, unlike MLL3/4, p300/CBP activity is tightly linked with both EWS::FLI1 binding and transcriptional output at the *NKX2-2* enhancer. The reduction of H3K27ac was also apparent at a later timepoint (seven days) following EWS::FLI1 genetic deletion (Fig. EV2J). Similarly, we did observe a decrease in H3K4me1 at the *NKX2-2* enhancer following EWS::FLI1 deletion, suggesting that loss of H3K4me1 is a potential indirect and secondary effect to prolonged loss of EWS::FLI1 binding (Fig. EV2J).

To assess how rapid EWS::FLI1 degradation affected protein complex binding and histone modifications more broadly at EWS::FLI1 enhancers, we performed ChIP-seq 24 h post EWS::FLI1 degradation. As a control we performed EWS::FLI1 ChIP-seq in both degrader systems and observed a reduction in EWS::FLI1 binding at all enhancers regions, irrespective of whether the enhancer was associated with a downregulated gene (Fig. 2F). We observed a significant reduction in p300 binding and H3K27ac levels at sensitive enhancers following 24 h EWS::FLI1 degradation (Figs. 2G,H and EV2K; Datasets EV2 and 3). We also observed a decrease in CBP binding at the *NKX2-2* enhancer in both degrader systems, indicating that both p300 and CBP are reliant upon EWS::FLI1 for maintained chromatin binding (Fig. EV2L). In direct contrast to this, we observed no significant difference in MLL4, and UTX, binding despite transcriptional downregulation (Figs. 2I,J and EV2M,N). Notably, we did observe a decrease in WDR5 binding at EWS::FLI1 enhancers regions (Fig. EV2M,N), suggesting other MLL complex components may be perturbed in this setting. In concordance with stable binding of MLL4 and UTX, we did not observe a significant decrease in H3K4me1 in the A673-dHALO and EWS-502 dTAG systems (Fig. 2I,J; Datasets EV2–4). In contrast, we observed a decrease at all sensitive enhancers in both

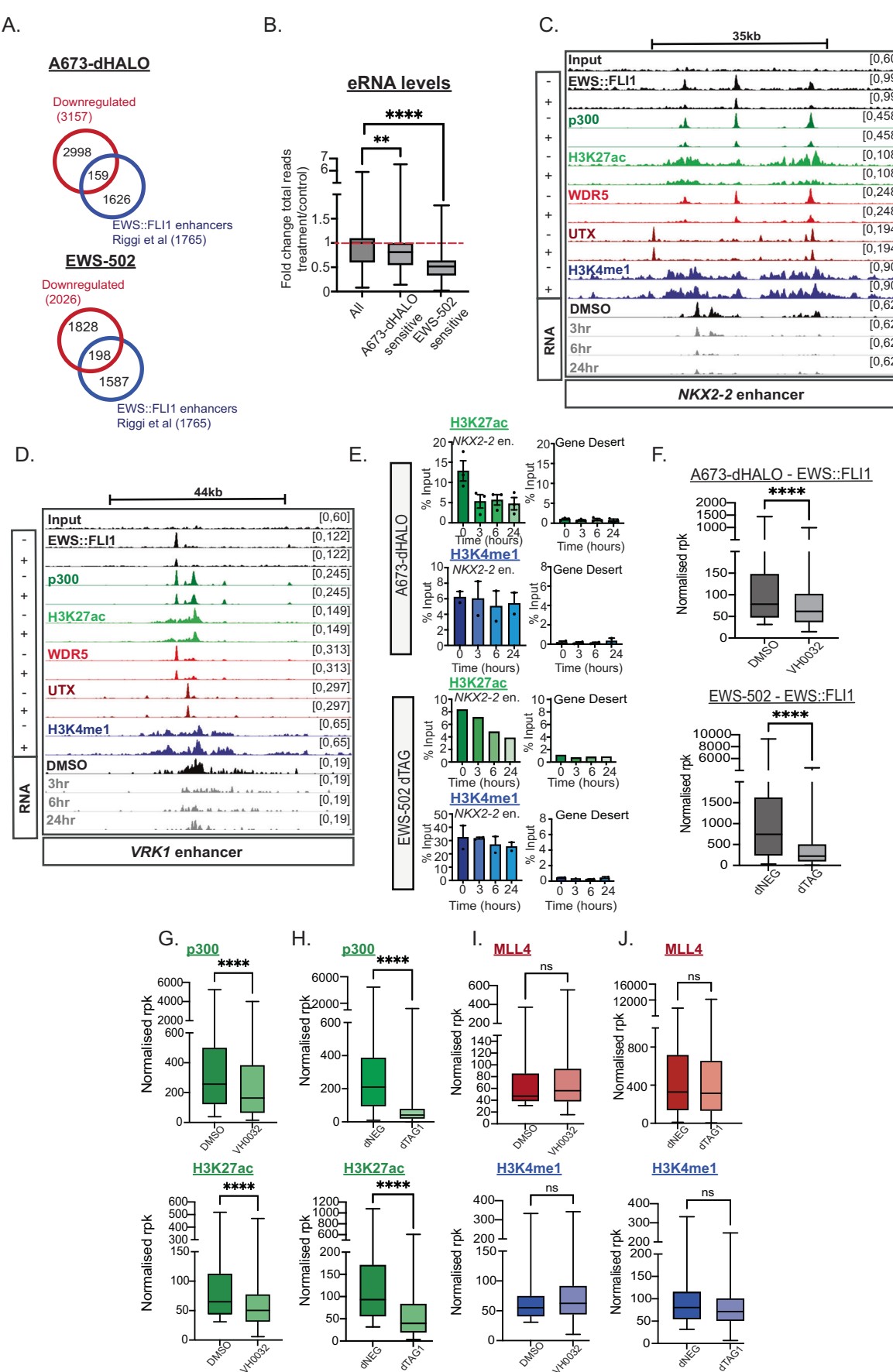

◀ **Figure 2.  Rapid degradation of EWS::FLI1 is acutely linked to reduced p300 occupancy and H3K27ac.**

(A) Overlap between downregulated genes in both A673-dHALO and EWS-502 dTAG system (log2FC <-0.5, FDR < 0.05) with 1785 EWS::FLI1 enhancers. (B) Fold change in eRNA signal at all EWS::FLI1 enhancers compared to sensitive EWS::FLI1 enhancers in the A673-dHALO and EWS-502 dTAG system. Kruskal–Wallis test, **$P = 0.0089$, ****$P < 0.0001$. Boxes depict the range between the first and third quartile. Central line within box shows the median value and whiskers highlight the maximum to minimum data points. eRNA signal calculated from three biological replicates. (C, D) ChIP-seq and RNA-seq tracks in A673-dHALO cells in DMSO control (-) and VH0032 ( + ) 24 h treatment at *NKX2-2* and *VRK1* enhancer. (E) H3K27ac (green) and H3K4me1 (blue) ChIP-qPCR at *NKX2-2* enhancer and gene desert control in both degradation systems. Error bars represent standard error of the mean from at least three biological replicates. Replicate 1 shown for H3K27ac in EWS-502 system. Replicate 2 in Fig. EV2I. (F) FLI1 ChIP-seq read normalized rpk at EWS::FLI1-sensitive enhancers in control and degrader conditions. Mann–Whitney *U* test, ****$P < 0.0001$. Boxes depict the range between the first and third quartile. Central line within box shows the median value and whiskers highlight the maximum to minimum data points. ChIP-seq from one biological replicate. (G) Normalized rpk for p300 and H3K27ac ChIP-seq at EWS::FLI1-sensitive enhancers in A673-dHALO system. Mann–Whitney *U* test, ****$P < 0.0001$. Boxes depict the range between the first and third quartile. Central line within box shows the median value and whiskers highlight the maximum to minimum data points. ChIP-seq from one biological replicate. (H) Normalized rpk for p300 and H3K27ac ChIP-seq at EWS::FLI1-sensitive enhancers in EWS-502 dTAG system. Mann–Whitney *U* test, ****$P < 0.0001$. Boxes depict the range between the first and third quartile. Central line within box shows the median value and whiskers highlight the maximum to minimum data points. ChIP-seq from one biological replicate. (I) Normalized rpk for MLL4 and H3K4me1 ChIP-seq at EWS::FLI1-sensitive enhancers in A673-dHALO system. Mann–Whitney *U* test, ns $P = 0.997$, 0.0738, respectively. Boxes depict the range between the first and third quartile. Central line within box shows the median value and whiskers highlight the maximum to minimum data points. ChIP-seq from one biological replicate. (J) Normalized rpk for MLL4 and H3K4me1 ChIP-seq at EWS::FLI1-sensitive enhancers in EWS-502 dTAG system (right). Mann–Whitney U test, ns $P = 0.275$, 0.6791, respectively. Boxes depict the range between the first and third quartile. Central line within box shows the median value and whiskers highlight the maximum to minimum data points. ChIP-seq from one biological replicate. Source data are available online for this figure.

H3K4me1 and H3K27ac following EWS::FLI1 genetic deletion (Fig. EV2O; Dataset EV4). These data demonstrate that p300/CBP binding and H3K27ac levels are dependent upon EWS::FLI1 occupancy at EWS::FLI1-sensitive enhancers, which occurs almost immediately following loss of EWS::FLI1 binding itself.

## Loss of p300/CBP and MLL3/4 protein function alter the EWS::FLI1 enhancer histone modification landscape

p300/CBP and H3K27ac levels are highly dependent upon EWS::FLI1 binding at enhancers in comparison to MLL4 and H3K4me1. Next, we wanted to determine the extent by which these two major chromatin complexes actively contribute to the regulation of EWS::FLI1 enhancer activity and gene regulation. To do this, we aimed to compare the effects of independently perturbing both p300/CBP and MLL3/4 function.

To investigate p300/CBP function, we used two distinct molecular tools targeting both p300 and CBP function. First, we used the potent p300/CBP degrader dCBP-1 to rapidly and selectively degrade both proteins (Vannam et al, 2021). We compared this to catalytic inhibition of p300/CBP using A-485, a p300/CBP acetyltransferase inhibitor (Lasko et al, 2017). We found that both p300 and CBP were globally depleted after 6 h of 100 nM dCBP-1 treatment, accompanied by reductions in bulk levels of H3K27ac and H3K18ac (Fig. 3A). Consistent with the intrinsic hook effect of PROTAC molecules, we observed a rebound of p300/CBP and associated histone modifications at higher concentrations (Fig. 3A) (Pettersson and Crews, 2019). Reduced p300 chromatin binding was verified by ChIP-qPCR (Fig. EV3A). Similarly, six hours of A-485 treatment led to a decrease in p300/CBP products H3K18ac and H3K27ac (Fig. 3B), while protein levels of p300/CBP remained stable (Fig. EV3B).

We next examined how histone acetylation is affected at EWS::FLI1 enhancers following p300/CBP perturbation. Traditionally, p300/CBP has been associated with H3K27ac and enhancer activity (Creyghton et al, 2010). However, recent studies have shown that additional acetylation marks, including H3K18ac and acetylated residues, are also dynamically linked to p300/CBP function and active enhancers (Weinert et al, 2018; Narita et al, 2023). Therefore, we

expanded our analysis to include these post-translational modifications. We performed ChIP-seq and ChIP-qPCR for H3K27ac, H3K18ac and H2BK20ac following dCBP-1 treatment in SKNMC, A673 and EWS-502 cells (Figs. 3C and EV3C–F; Dataset EV4). We observed a near-complete loss of these modifications at all EWS::FLI1 enhancers including the EWS::FLI1-sensitive subset including the *NKX2-2* enhancer. Comparatively, upon A-485 treatment, we observed an overall reduction in histone acetylation at all EWS::FLI1 enhancers (Figs. 3D–F and EV3G–J; Dataset EV4). Notably, this decrease in histone acetylation was more pronounced in the dCBP-1 setting compared to A-485 treatment (Fig. 3G,H) potentially indicative of mechanistic differences between p300/CBP catalytic inhibition versus total depletion of both proteins. We also observed no decrease in H3K4me1 following A-485 treatment, indicating that the catalytic activity of p300/CBP is not required for MLL3/4 methyltransferase function in this setting (Fig. 3I) (Lai et al, 2017).

To compare these p300/CBP-specific changes in the histone modification landscape to MLL3/4 activity, we performed genetic deletion of both MLL3 and MLL4 (MLL3/4 double knockout, DKO) in A673 and SKNMC cells. We confirmed near-loss of MLL3 and MLL4 protein levels seven days post editing, compared to a control gRNA targeting GFP (Fig. 3J). We confirmed that depletion of MLL3/4 protein levels was coupled with a decrease in bulk H3K4me1 levels consistent with MLL3/4 being major mammalian H3K4me1 methyltransferase proteins (Fig. 3J) (Lee et al, 2013). We performed ChIP-seq and ChIP-qPCR for histone modifications including H3K4 methylation (me1/2) in both A673 and SKNMC cells. In contrast to loss of p300/CBP activity, we observed an almost complete loss of H3K4me1 at EWS::FLI1 enhancers (Figs. 3K,L and EV3K–L; Dataset EV4). H3K4me2 was also greatly reduced (Fig. 3K,M). H3K27ac has been shown previously to be partially dependent on MLL3/4 activity in other cellular settings (Lai et al, 2017; Kang et al, 2021). In the context of EWS::FLI1 enhancers, we observed a reduction, but not total loss, of H3K27ac following MLL3/4 DKO (Figs. 3N and EV3N,O; Dataset EV4), suggesting that p300/CBP catalytic activity is modestly perturbed but still active. When we compared H3K4me1, H3K4me2 and H3K27ac at the subset of EWS::FLI1-sensitive enhancers, we

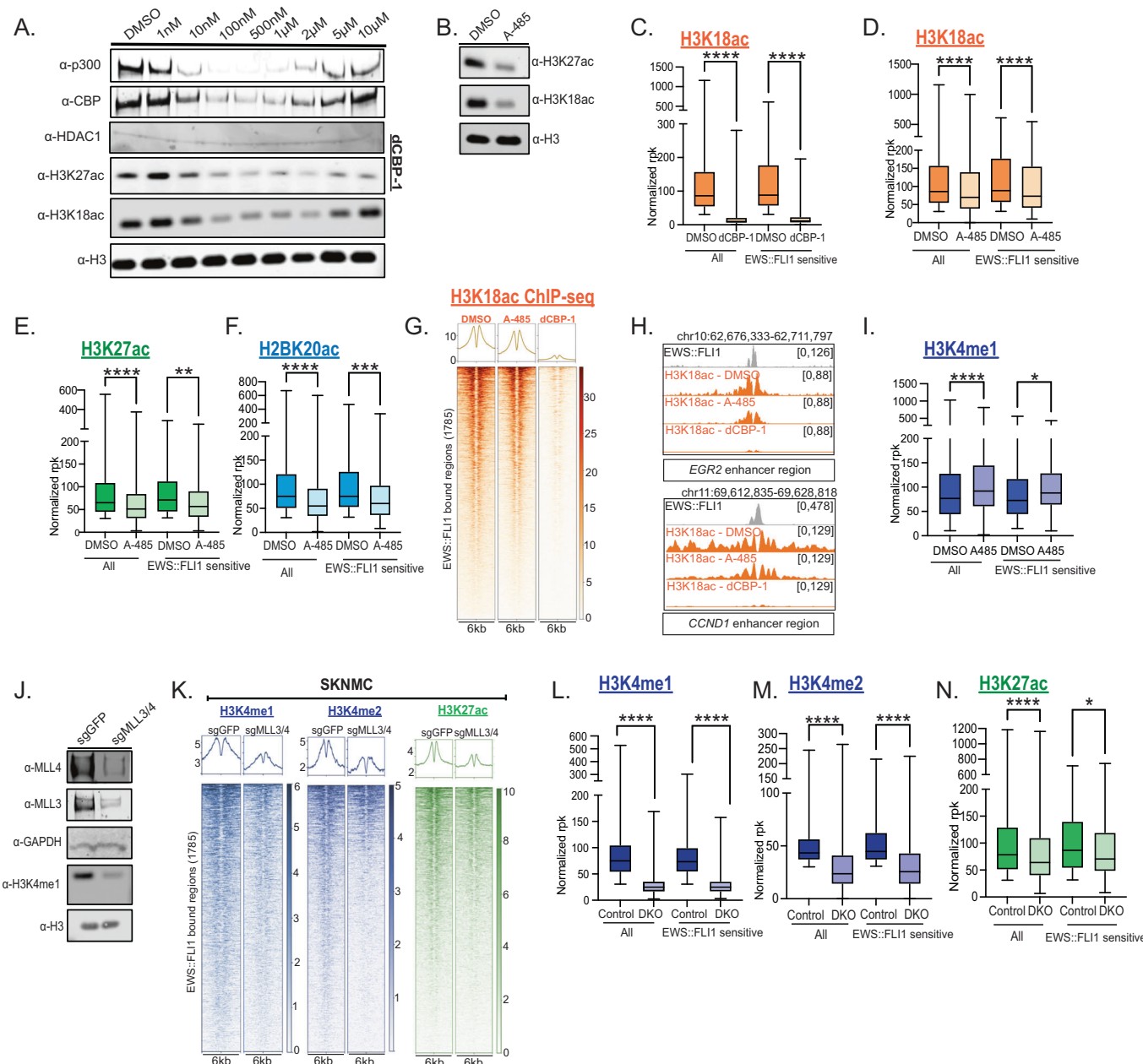

**Figure 3. Loss of p300/CBP and MLL3/4 protein function alter the EWS::FLI1 enhancer histone modification landscape.**

(A) p300, CBP, H3K27ac and H3K18ac western blot following nuclear extraction, and histone extraction, of SKNMC cells in DMSO control and 6 h 100 nM dCBP-1 treatment at different concentrations. (B) H3K27ac and H3K18ac western blot following histone extraction in 6 h 1 µM A-485 treated SKNMC cells. (C) H3K18ac normalized rpk at all EWS::FLI1 enhancers compared to EWS::FLI1-sensitive enhancers in DMSO and dCBP-1 (6 h) condition in SKNMC cells. Mann–Whitney *U* test, ****P < 0.0001. Boxes depict the range between the first and third quartile. Central line within box shows the median value and whiskers highlight the maximum to minimum data points. ChIP-seq from one biological replicate. (D–F) Histone modification ChIP-seq normalized rpk at all EWS::FLI1 enhancers and sensitive enhancers in DMSO control and A-485 (6 h) condition in SKNMC cells. Mann–Whitney *U* test, **P = 0.089, ***P = 0.004, ****P < 0.0001. Boxes depict the range between the first and third quartile. Central line within box shows the median value and whiskers highlight the maximum to minimum data points. ChIP-seq from one biological replicate. (G) Tornado plots showing H3K18ac ChIP-seq signal at EWS::FLI1 enhancers in DMSO, A-485 and dCBP-1 conditions in SKNMC cells. (H) Example ChIP-seq tracks at *EGR2* and *CCND1* enhancers for H3K18ac in DMSO, A-485 and dCBP-1 conditions in SKNMC cells. (I) H3K4me1 normalized rpk at all EWS::FLI1 enhancers compared to sensitive enhancers in DMSO and A-485 condition in SKNMC cells. Mann–Whitney *U* test, ****P < 0.0001, *P = 0.0113. Boxes depict the range between the first and third quartile. Central line within box shows the median value and whiskers highlight the maximum to minimum data points. (J) MLL3, MLL4 and H3K4me1 western blot following nuclear extract and histone extraction in sgGFP control and sgMLL3/4 DKO A673 cells. (K) Tornado plots showing H3K4me1, H3K4me2 and H3K27ac ChIP-seq signal at EWS::FLI1 enhancers in sgGFP control and sgMLL3/4 DKO setting in SKNMC cells (L–N) H3K4me1, H3K4me2 and H3K27ac normalized rpk at all EWS::FLI1 enhancers compared to sensitive enhancers in sgGFP control and sgMLL3/4 DKO setting. Mann–Whitney *U* test, *P = 0.0421, ****P < 0.0001. Boxes depict the range between the first and third quartile. Central line within box shows the median value and whiskers highlight the maximum to minimum data points. ChIP-seq from one biological replicate. Source data are available online for this figure.

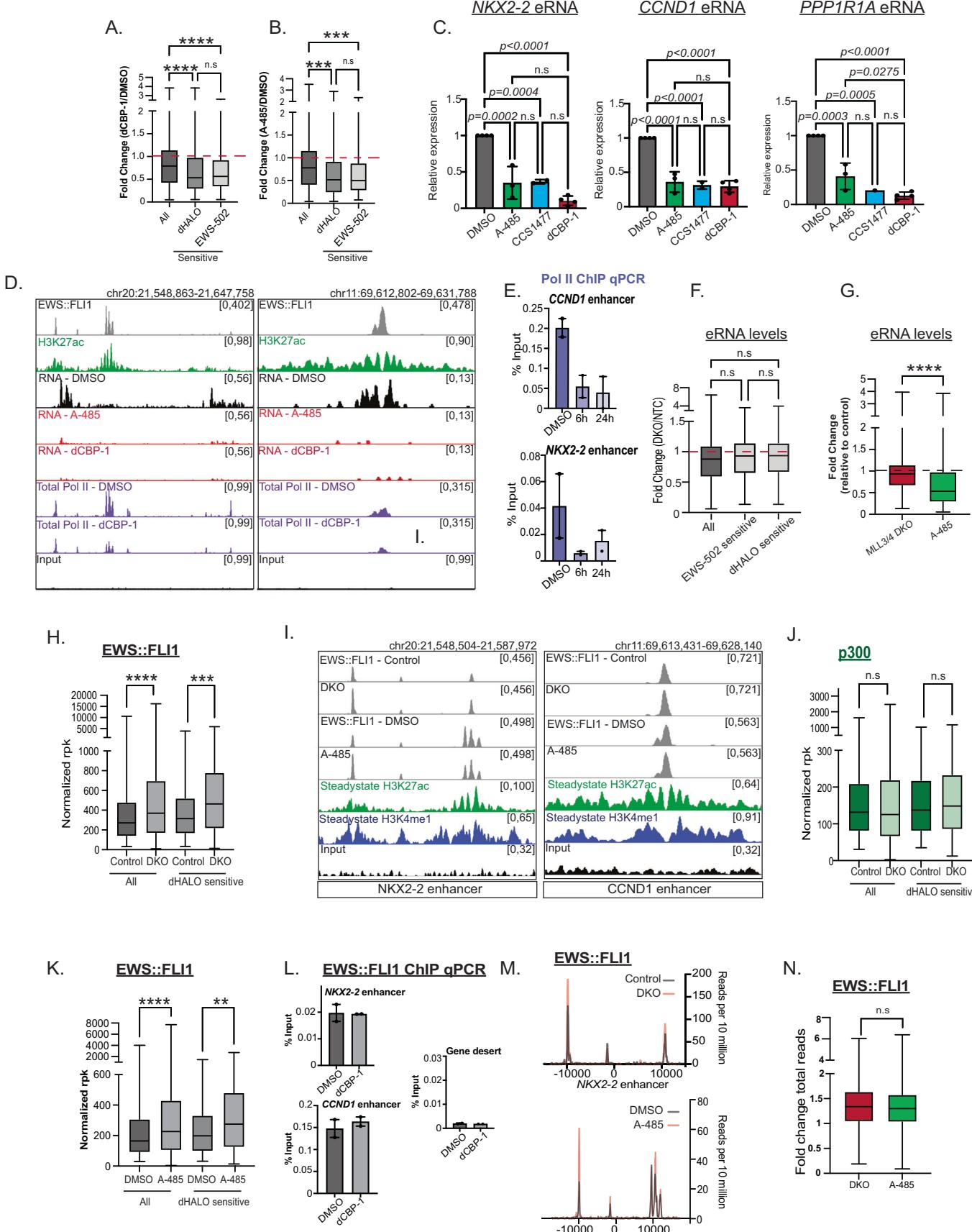

**Figure 4. p300/CBP is required for enhancer RNA levels and RNA polymerase II occupancy at EWS::FLI1 enhancers independently of EWS::FLI1 binding.**

(A, B) eRNA fold change at all EWS::FLI1 enhancers and EWS::FLI1-sensitive enhancers in dCBP-1 (6 h) and A-485 (6 h) treatment compared to DMSO control. Kruskal–Wallis test, (dCBP-1; ****$P < 0.0001$, A-485; ***$P = 0.0608$, $P = 0.0006$ ns, $P = 0.99$). Boxes depict the range between the first and third quartile. Central line within box shows the median value and whiskers highlight the maximum to minimum data points. eRNA signal calculated from three biological replicates, (C) RT-qPCR for eRNA of NKX2-2, CCND1 and PPP1R1A enhancers following 6 h treatment of A-485, CCS1477, and dCBP-1 treatment in SKNMC cells. Error bars represent standard error of the mean from at least three biological replicates. Kruskal–Wallis test, exact $P$ values indicated on figure. (D) Example tracks at NKX2-2 and CCND1 enhancers. (E) RNA polymerase II ChIP-qPCR at NKX2-2 enhancer in SKNMC cells following 6 h dCBP-1 treatment from two biological replicates. (F) Fold change in eRNA RNA-seq reads at all EWS::FLI1 enhancers and at sensitive enhancers from both degrader settings following MLL3/4 DKO. Mann–Whitney $U$ test, ns $P > 0.01$. Boxes depict the range between the first and third quartile. Central line within box shows the median value and whiskers highlight the maximum to minimum data points. eRNA signal calculated from three biological replicates. (G) Fold change in eRNA RNA-seq reads at all EWS::FLI1 enhancers and at sensitive enhancers from both degrader settings following MLL3/4 DKO compared to A-485 treatment. Mann–Whitney $U$ test, ****$P < 0.0001$. Boxes depict the range between the first and third quartile. Central line within box shows the median value and whiskers highlight the maximum to minimum data points. eRNA signal calculated from three biological replicates. (H) EWS::FLI1 normalized rpk at all EWS::FLI1 enhancers compared to sensitive enhancers in sgGFP control and sgMLL3/4 DKO. Mann–Whitney $U$ test, ***$P = 0002$, ****$P < 0.0001$. Boxes depict the range between the first and third quartile. Central line within box shows the median value and whiskers highlight the maximum to minimum data points. ChIP-seq from one biological replicate. (I) Example tracks at NKX2-2 and CCND1 enhancers. (J) p300 normalized rpk at all EWS::FLI1 enhancers compared to sensitive enhancers in sgGFP control and sgMLL3/4 DKO. Mann–Whitney $U$ test, ns $P > 0.01$. Boxes depict the range between the first and third quartile. Central line within box shows the median value and whiskers highlight the maximum to minimum data points. ChIP-seq from one biological replicate. (K) EWS::FLI1 normalized rpk at all EWS::FLI1 enhancers compared to sensitive enhancers in DMSO and. A-485 conditions in SKNMC cells. Mann–Whitney $U$ test, **$P = 0.021$, ****$P < 0.0001$. Boxes depict the range between the first and third quartile. Central line within box shows the median value and whiskers highlight the maximum to minimum data points. ChIP-seq from one biological replicate. (L) FLI1 ChIP-qPCR in DMSO control and dCBP-1 condition in SKNMC cells from two biological replicates. (M) Histograms showing EWS::FLI1 (Black—Control and Red—DKO/A-485) ChIP-seq reads per 10 million ($Y$ axis) at NKX2-2 enhancer. (N) Fold change in EWS::FLI1 ChIP-seq signal at all EWS::FLI1 enhancers following MLL3/4 DKO and A-485 treatment. Mann–Whitney $U$ test, ns $P > 0.01$. Boxes depict the range between the first and third quartile. Central line within box shows the median value and whiskers highlight the maximum to minimum data points. ChIP-seq from one biological replicate. Source data are available online for this figure.

detected no pronounced perturbation following MLL3/4 DKO (Fig. 3L–N).

## p300/CBP are required for enhancer RNA levels and RNA polymerase II occupancy independent of EWS::FLI1 binding

Having observed that targeting p300/CBP as well as MLL3/4 disrupts the histone modification landscape of active EWS::FLI1 enhancers, we aimed to evaluate the effect on enhancer activity by measuring eRNA levels at all EWS::FLI1 enhancer regions compared to sensitive enhancers. Strikingly, following dCBP-1 and A-485 treatment, we detected a significant decrease in eRNA at EWS::FLI1-sensitive enhancers, indicating that enhancers most reliant upon EWS::FLI1 are also particularly sensitive to loss of p300/CBP function (Fig. 4A,B,D; Dataset EV1). Using RT-qPCR to quantify this perturbation at well-characterized enhancers of NKX2-2, CCND1 and PPP1R1A, we compared dCBP-1 and A-485 settings to CCS1477 treatment; a p300/CBP bromodomain inhibitor (Nicosia et al, 2023; Welti et al, 2021). Here we observed a significant, comparable decrease in eRNA levels across all three conditions (Welti et al, 2021; Nicosia et al, 2023) (Fig. 4C). This highlights the likely multi-functional role p300/CBP plays not just in acetylation but as important scaffolding proteins at these pathogenic regions.

To evaluate the non-catalytic role of p300/CBP in eRNA synthesis, relative to histone acetylation at EWS::FLI1 enhancers, we pre-treated SKNMC cells with the pan histone deacetylase inhibitor SAHA for 24 h. This was to ensure maximal inhibition before subsequent treatment of dCBP-1 for 6 h. We compared this to dCBP-1 and SAHA treatment alone. SAHA treatment alone led to a slight elevation in bulk histone acetylation, while dCBP-1 alone reduced it (Fig. EV4A). Maintenance of histone acetylation levels was observed in the combined treatment along with loss of p300/CBP proteins (Fig. EV4A). Bulk EWS::FLI1 protein levels were slightly decreased with SAHA and the combined treatment, but ChIP-qPCR at the CCND1 enhancer showed unchanged EWS::FLI1 occupancy, indicating stable enhancer binding (Fig. EV4B). On the

chromatin level, H3K27ac ChIP-seq confirmed that dCBP-1 reduced acetylation at EWS::FLI1 enhancers while SAHA alone or combination treatment preserved it (Fig. EV4C). To assess enhancer activity, we measured eRNA levels at NKX2-2 and CCND1 enhancers. dCBP-1 treatment led to a decrease in eRNA levels in contrast with an increase above basal level following SAHA treatment (Fig. EV4C). Combination treatment led to a partially rescued eRNA level, suggesting that both catalytic and non-catalytic functions of p300/CBP are required to maintain full EWS::FLI1 enhancer activity (Fig. EV4D).

In concordance with eRNA perturbation following dCBP-1 and A-485 treatment, we also observed reduced total RNA polymerase II occupancy at EWS::FLI1 enhancer regions suggesting that loss of p300/CBP activity is sufficient to destabilize RNA polymerase II from EWS::FLI1-regulated enhancers (Figs. 4D,E and EV4E). These results contrasted with MLL3/4 DKO where we observed no significant difference in eRNA levels at EWS::FLI1 enhancers indicating that MLL3/4 play a less prominent role in the regulation of transcriptional activity at EWS::FLI1 enhancer regions (Figs. 4F,G and EV4F).

As loss of MLL3/4 proteins do not seem to perturb EWS::FLI1-driven enhancer activity, we hypothesized that the stability of EWS::FLI1 at chromatin, along with sustained p300/CBP activity, may partially account for maintained enhancer activity in the absence of MLL3/4. To test this, we performed EWS::FLI1 and p300 ChIP-seq in sgGFP control and MLL3/4 DKO conditions in A673 cells. When examining all EWS::FLI1 enhancers together, we observed a modest increase in EWS::FLI1 occupancy (Fig. 4H,I; Dataset EV4). Similarly, p300 binding was maintained on chromatin at EWS::FLI1 enhancers in the absence of MLL3/4 (Figs. 4I,J and EV4G; Dataset EV4). Taken together, this data suggests that the stabilization of both EWS::FLI1 and p300 at EWS::FLI1 enhancers is not dependent upon MLL3/4 occupancy and this may be partly responsible for maintained enhancer activity in the absence of MLL3/4.

As loss of p300/CBP function perturbs EWS::FLI1 enhancer activity, we wanted to assess the impact on the onco-fusion protein following p300/CBP targeting. Importantly, we observed no

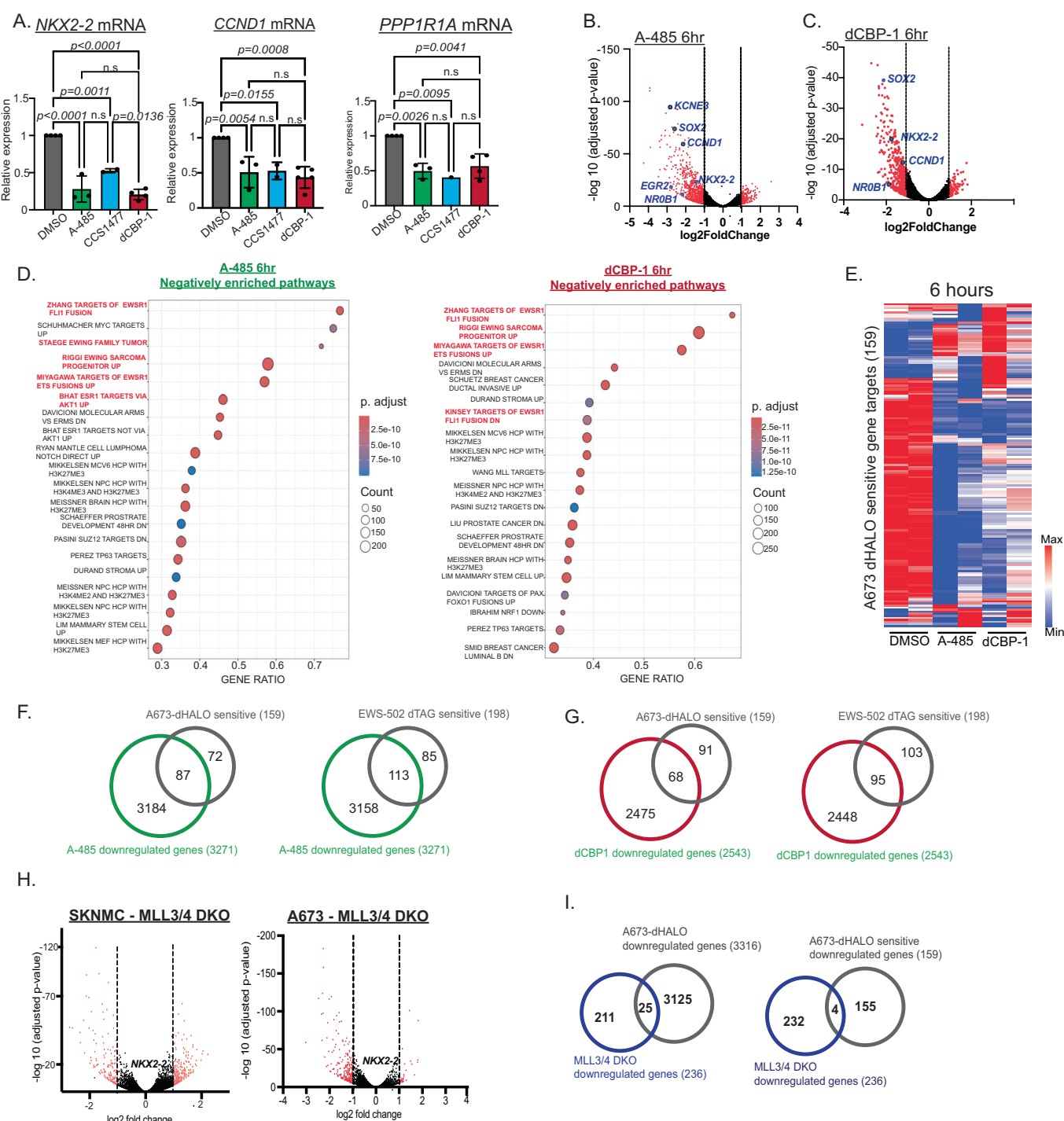

reduction in total EWS::FLI1 protein levels in either dCBP-1 or A-485 treated cells (Fig. EV4H). Furthermore, in both the dCBP-1 and A-485 conditions, we observed sustained EWS:FLI1 occupancy at enhancer regions by ChIP-seq and qPCR (Figs. 4K,L and EV4I). Interestingly, we observed a slight increase in EWS::FLI1 binding following A-485 treatment at EWS::FLI1 enhancer regions, including *NKX2-2* and *CCND1* in a similar manner to loss of MLL3/4 proteins (Fig. 4M,N; Dataset EV4). This indicates that p300/CBP activity is critical for the regulation of eRNA synthesis

and RNA polymerase II occupancy at enhancer independent of EWS::FLI1 binding and these enhancers require both EWS::FLI1 and p300/CBP to maintain baseline activity.

## p300/CBP are indispensable for EWS::FLI1-regulated gene expression

As we observed that p300/CBP are required to maintain an active EWS::FLI1-regulated enhancer landscape, we wanted to assess how

**Figure 5. p300/CBP function are central to the EWS::FLI1 gene regulatory network.**

(A) RT-qPCR for mRNA of *NKX2-2, CCND1* and *PPP1R1A* genes following 6 h treatment of A-485, CCS1477 and dCBP-1 treatment in SKNMC cells. Error bars represent standard error of the mean from at least three biological replicates. Kruskal–Wallis test, exact *P* values indicated on figure. (B, C) Volcano plot depicting differential gene expression following 6 h A-485 and dCBP-1 treatment. Red dots and dashed lines represent genes with a log2FC of >−/+1. Gene expression data calculated from three biological replicates. Significant values measured using a negative binomial wald test. (D) Gene set enrichment scores showing top 20 negatively enriched gene sets following 6 h A-485 and 6 h dCBP-1 treatment, in order of enrichment in SKNMC cells. Color represents *P*.adjust values from a Benjamini-Hochberg (BH) procedure test. Gene ratio represents total expressed genes in given gene set. (E) Z-scores of A673-sensitive EWS::FLI1 genes in DMSO, dCBP-1 and A-485 treatment conditions. (F) Overlap between downregulated A673-dHALO and EWS-502 dTAG-sensitive genes and genes downregulated following A-485 treatment (6 h). (G) Overlap between downregulated A673-dHALO and EWS-502 dTAG-sensitive genes and genes downregulated following dCBP-1 treatment (6 h). (H) Volcano plot depicting differential gene expression following 7 days of MLL3/4 DKO in A673 cells. Red dots and dashed lines represent genes with a log2FC of <−/+1, FDR < 0.05. Significant values measured using a negative binomial wald test Gene expression data calculated from three biological replicates. (I) Overlap between all, and sensitive, downregulated EWS::FLI1-sensitive genes and genes downregulated following MLL3/4 DKO (6 h). Source data are available online for this figure.

this disruption in activity affects downstream EWS::FLI1 gene target expression. To assess whether p300/CBP function is critical for EWS::FLI1 gene regulation, we performed RT-qPCR at well-studied EWS::FLI1 gene targets of *NKX2-2, CCND1* and *PPP1R1A* after six hours of A-485, CCS1477 and dCBP-1 treatment. In corroboration with eRNA RT-qPCR at the corresponding enhancers, we observed a significant decrease in expression of these EWS::FLI1 gene targets using all three small molecules compared to the DMSO control (Fig. 5A). This prompted us to assess gene expression changes on a more global scale. RNA-seq following 6 h of A-485 and dCBP-1 treatment in SKNMC and EWS-502 cells revealed many downregulated genes in both settings including some of the most well-characterized EWS::FLI1 gene targets (Fig. 5B,C; Datasets EV6 and 7). When comparing downregulated genes between the two conditions, we noted a substantial overlap, with over 80% of the genes downregulated by dCBP-1 treatment also exhibiting decreased expression with A-485 treatment (Fig. EV5A). To determine whether EWS::FLI1 regulated pathways are particularly enriched within the downregulated genes following A-485 and dCBP-1 treatment, we performed GSEA analysis. Strikingly, we observed that many EWS::FLI1 regulated gene signatures were significantly enriched among the top 20 negatively enriched pathways (Figs. 5D and EV5B,C). Interestingly, we also observed enrichment of H3K27me3 and polycomb repression signatures, possibly indicating a disturbance in the balance between H3K27ac and H3K27me3 (Fig. 5D) (Pasini et al, 2010). When we specifically examined the EWS::FLI1-sensitive genes identified in both the A673-dHALO and EWS-502 dTAG systems, we observed a large proportion of downregulated genes following A-485 and dCBP-1 treatment (Figs. 5E and EV5D). Specifically, 54% of the A673-dHALO- and 57% of EWS-502 dTAG-sensitive genes were downregulated following A-485 treatment (Fig. 5F). 42% and 47% were downregulated in both systems following dCBP-1 treatment, respectively (Fig. 5G). Thus, highlighting a major role of p300/CBP in regulating EWS::FLI1 enhancers and associated gene targets.

To further distinguish the effects on gene expression following loss of histone acetylation compared to p300/CBP proteins, we again compared dCBP-1 and SAHA treatment. We observed, in a similar manner to eRNA expression, that combinatorial treatment using SAHA and dCBP-1 led to a partial rescue of gene expression (Fig. EV5E), suggesting that maintenance of acetylation levels in the absence of p300/CBP is sufficient to rescue downstream gene transcription to an intermediate level. To compare the gene regulatory role of MLL3/4 in Ewing sarcoma to that of p300/CBP, we performed RNA-seq in SKNMC and A673 cells seven days post

CRISPR editing. We observed 236 and 160 downregulated genes in the A673 and SKNMC cells, respectively (Fig. 5H; Dataset EV5). Very few upregulated genes were observed in both settings consistent with MLL3/4 supporting active transcription (Fig. 5H). To assess how many of these genes are involved in EWS::FLI1 gene regulation, we overlapped MLL3/4 DKO downregulated genes and those downregulated in the A673-dHALO system. We observed very few common downregulated genes between MLL3/4 DKO and those downregulated following EWS::FLI1 degradation (Fig. 5I). Furthermore, the number of co-regulated genes were even fewer, with only 4 (0.6%) and 2 (1.2%) genes in A673 and SKNMC, respectively, when we compared MLL3/4 DKO downregulated genes with the 159 A673-dHALO EWS::FLI1-sensitive genes (Figs. 5I and EV5F). Even when comparing downregulated genes in the MLL3/4 DKO and EWS::FLI1 KO condition (performed at much later timepoint post perturbation) we observed minimal overlap (Fig. EV5G). Interestingly, we did observe some overlap between MLL3/4 DKO downregulated genes following A-485 inhibition (Fig. EV5H). These genes were not co-regulated by EWS::FLI1 and potentially reflect the regulation of non-EWS::FLI1 controlled enhancers. This data highlights that MLL3/4 regulate an EWS::FLI1 independent gene expression network in Ewing sarcoma and do not hold a major role in the regulation of genes driven by the oncogenic fusion protein. This is in stark contrast to p300/CBP which play a central role in the regulation of EWS::FLI1-mediated gene regulation.

## p300/CBP are critical for Ewing sarcoma tumor growth in vivo

Perturbation of p300/CBP function leads to downregulation of EWS::FLI1 gene target expression and reduced enhancer activity. Therefore, we hypothesized that Ewing sarcoma cell growth would be affected due to this dysregulation of critical gene target expression. To test this in an in vitro setting, we performed drug titration assays using A-485 and CCS1477 as single agent and dual treatments in A673 and SKNMC cells. We observed that both cell lines exhibited similar sensitivity to both A-485 and CCS1477 treatment with IC50 values in the low μM range after 6 days of treatment (Fig. 6A). The level of sensitivity observed aligns with other cancer models sensitive to p300/CBP inhibition (Ji et al, 2022; Lasko et al, 2017). When combined, the sensitivity of treatment increased further (Fig. 6A). We further evaluated the combined effect of both inhibitors across all cell lines (A673, SKNMC, and EWS-502) using SynergyFinder 3.0. Since most synergy scores

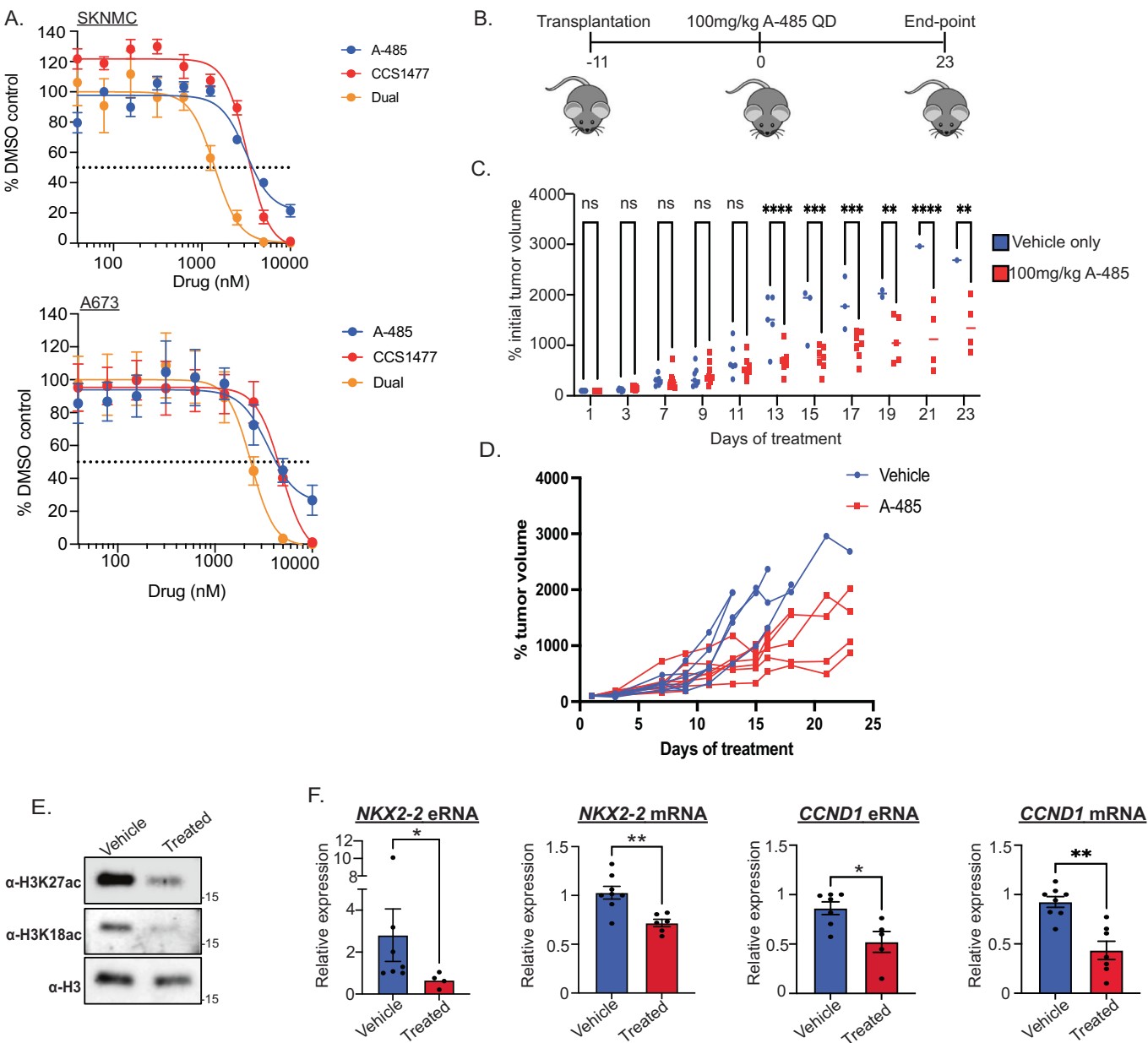

**Figure 6. p3OO/CBP inhibition slows Ewing sarcoma tumor growth in vivo.**

(A) Cell viability of SKNMC and A673 cells treated with A-485, CCS1477 and Dual combination for 6 days measured by CellTiter-Glo. Error bars represent standard deviation of the mean from three biological replicates. (B) Schematic showing in vivo experimental method. (C) Graph showing percentage initial tumor volume of control mice (blue) and treated (red) for 23 days of A-485 treatment. Each point represents an individual mouse with at least four mice per condition at the end-point. Two-way ANOVA, **P = 0.0024, 0.0010, ***P = 0.0001, 0.0007, ****P < 0.0001. (D) Spaghetti plot showing percentage initial tumor volume of individual mice in control (blue) and treated (red) conditions. (E) Western blot analysis of H3K27ac and H3K18ac following histone extraction of tumor cells extracted from a vehicle-only and a A-485-treated mouse. (F) RT-qPCR at NKX2-2 and CCND1 eRNA and mRNA in vehicle (blue) and treated (red) mice. Each individual point represents expression from a single animal. Error bars represent standard deviation of the mean from at least four biological replicates. Mann–Whitney U test, *P = 0.00242, 0.029, **P = 0.0073, 0.0012. Source data are available online for this figure.

ranged between 0 and 10, we concluded that the enhanced sensitivity to the combined A-485 and dCBP-1 treatment is likely additive (Fig. EV6A). Taken together, these data suggest that multiple p300/CBP domains, including both the enzymatic domain and non-enzymatic bromodomain, are likely important for Ewing sarcoma cell growth.

To further our understanding of how p300/CBP inhibition might influence Ewing sarcoma growth in a physiologically relevant setting, we established an in vivo SKNMC xenograft model. Daily IP injection was performed with a vehicle control formulation or 100 mg/kg A-485 (Fig. 6B). We did not observe any dose-limiting toxicities under treatment, as exemplified by stable body weights

throughout the course of treatment (Fig. EV6B). Tumor volume was measured every other day and revealed a significant deceleration of tumor growth in mice treated with A-485 (Fig. 6C,D). To validate on-target efficacy in vivo, we performed histone extraction and western blotting for both H3K18ac and H3K27ac in vehicle-only and A-485-treated mice (Fig. 6E). We observed a global loss of both modifications, demonstrating that A-485 was permeable to the tumor tissue and inhibited p300/CBP enzymatic activity. Furthermore, we isolated RNA from at least six tumors in both conditions and performed RT-qPCR for *NKX2-2* and *CCND1* eRNA and mRNA. Strikingly, we observed a significant decrease in expression levels in A-485-treated tumors compared to vehicle only at target loci (Fig. 6F). Taken together, this demonstrates both that A-485 has in vivo, on-target efficacy in an Ewing sarcoma xenograft model and that p300/CBP are critical modulators for Ewing sarcoma cell growth and enhancer activity. Moreover, this highlights a potential application of p300/CBP inhibition in targeting Ewing sarcoma in a clinical setting. Thus, we propose a critical role for p300/CBP, but not MLL3/4, in the maintenance of EWS::FLI1 enhancer activity and downstream targets, significantly impacting Ewing sarcoma cell growth.

# Discussion

In this study, we investigate the mechanisms by which chromatin-modifying complexes, MLL3/4 and p300/CBP, maintain an active EWS::FLI1 enhancer landscape and gene expression program. We find that EWS::FLI1-regulated gene expression remains unaffected upon loss of MLL3/4 function despite depletion of both H3K4me1 and H3K4me2 levels, and a modest decrease in H3K27ac at enhancers. This decrease does not disrupt gene target expression, and both EWS::FLI1 and p300 occupancy at EWS::FLI1 enhancers are retained. In contrast, our observations reveal a dependency on p300/CBP activity for the maintenance of EWS::FLI1 gene target expression. This dependence is associated with greater reductions in histone acetylation and disruption of eRNA synthesis at EWS::FLI1-regulated enhancers. This perturbation was particularly noticeable at EWS::FLI1-sensitive enhancers and gene targets which were also most sensitive to loss of EWS::FLI1 binding in the degrader cell lines.

Studies investigating MLL3/4 function in active enhancer maintenance across various cellular contexts have yielded diverse findings. In mouse embryonic stem cells, Mll3/4 were shown to play a role in fine tuning wild-type enhancer activity. Notably, the non-catalytic function of Mll3/4 was shown to play a greater role in maintaining eRNA synthesis and Pol II occupancy, whereas H3K4me1 has been implicated to play a role in enhancer-promoter interactions (Dorighi et al, 2017; Yan et al, 2018). Other studies have shown that loss of MLL3/4 function can disrupt active enhancer features without affecting downstream gene expression significantly, suggesting they play a minor role in gene regulation depending upon the cellular context (Boileau et al, 2023). Although these proteins were shown to have some redundant function, there have been examples of disease settings were MLL4 has been shown to be more critical, and vice versa (Santos et al, 2014; Lai et al, 2017; Zhu et al, 2023; Wang et al, 2016). In the context of Ewing sarcoma and the regulation of EWS::FLI1 enhancers, our findings reveal that perturbation of both MLL3 and MLL4 exert minimal effects on

enhancer activity and downstream gene expression. This suggests that other regulatory mechanisms at EWS::FLI1 enhancers may override the loss of MLL3/4. This could underscore the strong dependence of these pathogenic enhancer regions on EWS::FLI1 for both de novo activation and maintenance, even in the absence of other activating enhancer features. Interestingly, we observed a slight increase in EWS::FLI1 binding following MLL3/4 DKO (Fig. 4H,I). Previously, studies have demonstrated that loss of activating transcription factors has led to sites of increased chromatin accessibility (Xu et al, 2021; Friman et al, 2019). Something similar could be occurring at EWS::FLI1 enhancers upon MLL3/4 DKO, which may permit an increase in EWS::FLI1 binding affinity to DNA.

In addition to enhancer maintenance, MLL3/4 and H3K4me1 have been implicated in enhancer initiation and poising (Rada-Iglesias et al, 2011). While our study demonstrates that MLL3/4 activity is dispensable for maintaining EWS::FLI1 enhancer activity, we cannot discount its potential role in enhancer activation. Previous work has shown that EWS::FLI1 recruits the MLL3/4 complex and promotes H3K4me1 at de novo enhancers in MSCs, suggesting a possible requirement for MLL3/4 activity in this process (Riggi et al, 2014). Further investigation into the functions of chromatin-modifying complexes, including MLL3/4 and p300/CBP, in EWS::FLI1 enhancer initiation will yield valuable insight.

In contrast to MLL3/4, our observations underscore the central role of p300/CBP function in sustaining EWS::FLI1 enhancer activity and regulating downstream gene expression. Interestingly, our findings reveal a more pronounced perturbation of histone acetylation with dCBP-1 treatment compared to A-485. This could indicate that in the complete absence of p300/CBP (with dCBP-1 treatment) histone deacetylation occurs more rapidly. This is consistent with previous studies showing that HDAC activity plays an important role in Ewing sarcoma (Schmidt et al, 2021; El-Naggar et al, 2019).

A-485 treatment demonstrated a similar effect on EWS::FLI1 gene target expression compared to dCBP-1. This suggests that despite p300/CBP playing potential non-catalytic roles at enhancers, the acetyltransferase activity is important in this context. p300/CBP have been shown to acetylate many histone residues as well as a plethora of non-histone proteins which are likely found at active enhancer regions (Weinert et al, 2018). Therefore, it is possible that reduced acetylation of other substrates may contribute to reduced gene expression.

Importantly, p300 and CBP are distinct homologous proteins, and context-dependent functional differences have been observed in other cancer settings (Durbin et al, 2022; Martire et al, 2020). Our study focuses on the dual role of p300/CBP using tools that simultaneously target both proteins. We demonstrate that inhibiting both disrupts EWS::FLI1 target gene regulation and impairs Ewing sarcoma cell growth. Future studies aimed at distinguishing the specific and separate function of p300 and CBP in this context would provide valuable insights.

Taken together, this study underscores some of the precise dependencies of specific enhancer subsets, such as EWS::FLI1 enhancers, on distinct chromatin protein complexes. More broadly, this highlights the context-dependent nature of enhancer regulation, where certain chromatin complexes may be required for steady-state regulation (i.e., enhancer maintenance) while others, such as MLL3/4, may play a more important role in state

transitions (i.e., enhancer initiation). Recent studies have expanded our understanding of chromatin complex function in enhancer activity by associating proteins classically linked with transcriptional elongation at genes, such as DOT1L and NSD1, with enhancer regulation (Godfrey et al, 2019; Sun et al, 2023). This emerging evidence suggests that diverse chromatin-modifying complexes beyond canonical enhancer regulators contribute to the regulation of different enhancer subsets. Further exploration to uncover novel functions of chromatin-modifying complexes at enhancers may help elucidate distinct enhancer regulatory mechanisms, particularly in disease contexts like Ewing sarcoma, where enhancer activity directly influences disease pathology.

# Methods

## Reagents and tools table

| Reagent/resource | Reference or source | Identifier or catalog number |
| --- | --- | --- |
| **Experimental models** | | |
| A673 | ATCC | CRL-1598 |
| SKNMC | ATCC | HTB-10 |
| EWS-502 dTAG | Stegmaier Lab; Nabet et al, 2020 | N/A |
| A673-dHALO | Tijan Lab; Chong et al, 2022 | N/A |
| **Recombinant DNA** | | |
| N/A | | |
| **Antibodies** | | |
| P300 | Bethyl | A300-358A |
| CBP | Bethyl | A300-362A |
| MLL4 | Millipore | ABE1867 |
| MLL4 | Sigma | HPA035977 |
| MLL3 | CST | 53641S |
| FLI1 | Abcam | ab133485 |
| WDR5 | Bethyl | A302-429A |
| UTX | Bethyl | A302-374A |
| PTIP1 | Bethyl | A300-370A |
| RbBP5 | Bethyl | A300-109A |
| ASH2L | Bethyl | A300-489A |
| H3K4me1 | Abcam | ab8895 |
| H3K4me2 | Abcam | ab7766 |
| H3K4me3 | Abcam | ab8580 |
| H3K27ac | Abcam | ab4729 |
| H3K18ac | Abcam | ab1191 |
| H2BK20ac | Abcam | ab177430 |
| HDAC1 | Cell Signaling Technologies | 5356S |
| H3 | Abcam | ab1791 |
| GAPDH | Cell Signaling Technologies | 5174S |
| **Oligonucleotides and other sequence-based reagents** | | |
| qPCR primers | | Dataset EV8 |

| Reagent/resource | Reference or source | Identifier or catalog number |
| --- | --- | --- |
| **Chemicals, enzymes, and other reagents** | | |
| dTAGv-1 | R&D Systems | #6914/5 |
| dTAGv-1-NEG | R&D Systems | #6915/5 |
| VH032-PEG5-C6-Cl | MedChem Express | #HY-112495 |
| A-485 | Qi Lab | N/A |
| dCBP-1 | Ott Lab | N/A |
| CCS1477 | MedChem Express | #HY-111784 |
| Xtremegene-9 | Sigma-Aldrich | #6365809001 |
| Polybrene | Fisher Scientific | #TR1003G |
| CellTiter-Glo | Promega | #G7571 |
| High-Capacity cDNA RT kit | Life Technologies | #4374966 |
| Power SYBR Green PCR MasterMix | Life Technologies | #4367660 |
| Taqman | Life Technologies | #4369016 |
| RNeasy mini kit | Qiagen | #74106 |
| 16% Formaldehyde | Life Technologies | #28908 |
| Disuccinimidyl Glutarate | Fisher Scientific | #PI20593 |
| ThruPLEX DNA-seq kit | Takara Bio | #R400676 |
| Hydroxypropyl-β-cyclodextrin | Fisher Scientific | #AC297561000 |
| Matrigel | VWR | #47743-722 |
| NEBNext rRNA Depletion V2 kit | New England Biolabs | #E7405L |
| NEBNext Ultra II RNA library prep kit | New England Biolabs | #E7770L |
| **Software** | | |
| Morpheus tool | https://software.broadinstitute.org/morpheus | |
| SynergyFinder 3.0 | https://synergyfinder.fimm.fi/ | |
| **Other** | | |
| Nextseq 500 75-cycle high-output kit | Ilumina | # 20024906 |

## Cell lines and tissue culture

All human cell lines were sourced from ATCC or DSMZ. Ewing sarcoma cell lines A673 (ATCC, CRL-1598) and SKNMC (ATCC, HTB-10) cells were grown in RPMI media supplemented with 10% fetal bovine serum (FBS, Gibco) and 1% L-Glutamine (Gibco) and 1× penicillin/streptomycin (Gibco). Exogenous EWS-502 dTAG cell line (Nabet et al, 2020), was grown in RPMI with 10% fetal bovine serum (FBS, Gibco), 1% L-Glutamine (Gibco), 1× penicillin/streptomycin (Gibco) and 1 μg/μl Blastocidin (Life Technologies). A673-dHALO (Chong et al, 2022) cells and HEK293T cells were grown in DMEM supplemented with 10% FBS, 1% L-Glutamine

and 1× penicillin/streptomycin. Ewing sarcoma cell line TC71 cells (DSMZ, ACC5-16) were grown in IMDM supplemented with 10% FBS and 1% L-Glutamine and Pen-Strep. All cell lines were cultured at 37 °C with 5% $CO_2$ and tested frequently for mycoplasma infections. Cell lines were maintained by splitting one in three using trypsinization (Gibco, 0.25%) three times per week.

## EWS::FLI1 degrader experiments

Ewing sarcoma cell lines were plated out into 10 cm³ tissue culture-treated plates the day before treatment at 33% confluency. For the EWS::FLI1 degrader experiments, 1 μM dTAGv-1 (R&D Systems) or dTAGv-1-NEG (R&D Systems) compounds were added to the EWS-502 cells at a concentration of 1 μM for up to 24 h. For the A673-dHALO system, either 1 μM DMSO or VH032-PEG5-C6-Cl (Medchem Express) was added for up to 24 h.

## Small-molecule inhibitors and dCBP-1 experiments

Ewing sarcoma cell lines were plated out into 10-cm³ tissue culture-treated plates the day before treatment at 33% confluency 1 μM DMSO, 1 μM A-485 (Qi lab) or 1 μM CCS1477 (MedChem express, #HY-111784) were added to cells and collected 6 or 24 h later. For washout experiments, cells were treated with DMSO or drug for 6 h followed by media replacement, with DMSO or dCBP-1, and collected 24 h later. For dCBP-1 experiments, 100 nM DMSO or 100 nM dCBP-1 was added to cells and collected 6 or 24 h later for downstream applications.

## Plasmids and sgRNA cloning

The sgRNA-CAS9 expressing lentivirus LentiV2 CRISPR plasmids conferring either Puromycin or Blasticidin resistance were obtained from Addgene and were used for all CRISPR experiments detailed in this study. Briefly, LentiV2 CRISPR plasmids were digested using BsmBI-V2 restriction enzyme (NEB #R0739S) at 50 °C for 2 h. sgRNAs were annealed using a heat gradient (See Dataset EV8 for all sgRNA sequences used in this study). Plasmid and gRNA were ligated and transformed using GC5 competent cells (Genesee Scientific) and plated onto ampicillin agar plates. Sanger sequencing was used to determine the correct insert, and cultures were purified using the Midiprep kit (Zymo).

## CRISPR genetic deletions and transduction

For CRISPR deletion experiments, lentiviral supernatant was produced using 293T cell transfection of the plasmid of interest and packaging plasmids (pMD2G and PAX) using Xtremegene-9 DNA transfection reagent (Sigma). Viral supernatant was collected 48 h after transfection. A673 or SKNMC cells were transduced with lentiviral supernatant in 6-well dish formats. Briefly, 1 mL viral supernatant per target was added to the well plus 1 mL of RPMI and 6 μg/ml polybrene (Fisher Scientific). Cells were spinfected at 600 × $g$ for 1 h at 37 °C and then incubated overnight. Next day, media was replaced with fresh media and split up into 10-cm dishes if confluency required it. 48 h post infection, cells were treated with either 2 μg/mL puromycin or 1 μg/mL blasticidin (or both in the case of MLL3 (blast) and MLL4 (puro) double genetic deletion). Cells were grown for 7 days, at which point they were collected for downstream applications.

## CellTiter-Glo and drug synergy experiments

1000 cells were seeded per well in a 96-well tissue culture-treated dish 24 h prior to administering drug. CellTiter-Glo (CTG, Promega) experiments were performed at day 6 and day 12 of treatment with either A-485, CCS1477 or combined A-485 and CCS1477. In brief, cells were trypsinized and resuspended in 200 μL of appropriate media. 20 μL of cells plus 20 μL CellTiter-Glo reagent were added to each well and incubated at room temperature (RT) for 30 min with shaking. Cell and CTG mixture were transferred to a flat-bottom white microplate and measured on the Clariostar microplate reader. All experiments were conducted with technical triplicates for each biological replicate ($n = 3$). Media-only wells were included as blank controls. For drug synergy experiments, CTG readings were used as input for the synergy-finder web browser (https://synergyfinder.fimm.fi/). Synergy scores were calculated using the Bliss model of drug synergy scoring.

## Protein extraction and western blotting

For histone modification western blotting, histone acid extraction was performed. Briefly, cells were lysed in BC300 (20 mM Tris-HCl pH 8.0, 300 mM KCl, 5 mM EDTA, 20% glycerol) + 0.5% NP40 and protease inhibitor cocktail (PIC) for 30 min on ice. Lysed cells were spun down, supernatant discarded, and the pellet resuspended in 0.4 M HCl for 30 min on ice. For western blotting, histone extracts were quantified using BCA and diluted in LDS buffer (Life Tech, #NP0007) plus 1/100 B-Mercaptoethanol on a 12% Bis-Tris gel using MES buffer. H3 was used as a loading control in all cases.

For non-histone proteins, a nuclear extraction protocol was used in most cases. Briefly, cells were pelleted and resuspended in hypotonic buffer for 10 min (10 mM Tris, pH 7.5, 10 mM KCl, 1.5 mM $MgCl_2$). Nuclei were pelleted and resuspended in low salt buffer (20 mM Tris pH 7.5, 12.5% glycerol, 1.5 mM $MgCl_2$, 0.2 mM EDTA pH 8.0). High salt buffer was then added dropwise to resuspended nuclei (20 mM Tris pH 7.5, 12.5% glycerol, 1.5 mM $MgCl_2$, 0.2 mM EDTA pH 8.0, 1.2 M KCl) and incubated with vortexing for 25 min at 4 °C. Following, nuclei were spun down and supernatant collected for western blotting purposes. HDAC1 was used in all cases as a loading control. Alternatively, where necessary BC300 (as indicated) + 0.1% NP40 extraction was performed for salt soluble protein extraction in which case GAPDH was used as a loading control.

## RNA isolation and RT-qPCR

All cells collected for RNA isolation were maintained at no more than 70% confluency upon collection point to maintain exponential cell growth rates. RNA was isolated using a RNeasy mini extraction kit. Upon collection, cells were lysed immediately with buffer RLT. And B-mercaptoethanol. Between 1 and $2 \times 10^6$ cells were collected for each RNA sample. For cDNA production, RNA was quantified and a standardized amount up to 2 μg per sample was taken forward for cDNA synthesis (High-Capacity cDNA reverse transcription kit, Life Technologies) using random primers. For eRNA cDNA production, Reverse Transcriptase (RT) negative control was made each time to account for potential gDNA contamination. Following cDNA production, RT-qPCR was performed. See Dataset EV8 for the list of qPCR primers used in this study.

## RNA sequencing

For RNA sequencing, isolated RNA was run on the Aligent 4200 Tapestation system to verify intact, high-quality RNA and quantified using qubit (Thermo Fisher). In total, 1 µg of total RNA was used to make Illumina-compatible libraries using NEBNext rRNA Depletion V2 kit (NEB, E7405) to remove ribosomal RNA. Libraries were sequence by paired-end sequencing using a 75-cycle high-output NextSeq 550 kit (Illumina). An average of 50 million reads per sample was generated to detect low-expressing RNA produced at enhancer regions.

## Chromatin immunoprecipitation (ChIP)

Chromatin immunoprecipitation experiments were performed on Ewing sarcoma cell lines. Briefly, between 1 and $2 \times 10^7$ cells were collected and crosslinked with either 1% formaldehyde (FA) for 10 min (histone proteins) or 2 µM Disuccinimidyl Glutarate (DSG) for 30 min followed by 1% FA for a further 10 min (chromatin proteins and EWS::FLI1). Fixed cells were washed and lysed in ChIP lysis buffer (20 mM Tris pH 7.5, 300 mM NaCl, 2 mM EDTA, 0.5% NP40, 1% Triton X-100) for 30 min on ice and spun down 5000 rpm for 10 min at 4 °C. Cell pellet was resuspended in 1 mL ChIP sonication buffer (0.1% SDS, 0.5% N-lauroylsarcosine, 1% Triton X-100, 10 mM Tris-HCl pH 8, 100 mM NaCl, 1 mM EDTA) and sonicated using the Covaris E220. Sonicated chromatin was then spun down at 13,000 rpm for 10 min and the supernatant collected and quantified. Between 75 and 250 µg of chromatin was taken per sample, per IP. Beads and antibodies were pre-coupled for 2–3 h at 4 °C with rotation. 2 µg of antibody per histone ChIP and 5 µg per non-histone ChIP were used with 20 µL dynabeads A or G. Bead:antibody complexes and chromatin were incubated overnight with rotation at 4 °C. The next day, beads were washed 3× with ChIP wash buffer (50 mM Tris-HCl, pH 7.5, 500 mM LiCl, 1 mM EDTA, 1% NP40, 0.7% Na-deoxycholate) and 1× with TE + 50 mM NaCl. ChIP samples were then eluted in ChIP elution buffer (50 mM Tris pH 8.0, 10 mM EDTA, 1% SDS) and de-crosslinked overnight at 65 °C. ChIP samples were subject to RNase A and Proteinase K digestion and column-purified. ChIP experiments were analyzed using qPCR, relative to input. See Dataset EV8 for qPCR primers.

## ChIP-sequencing and normalization

ChIP-seq libraries were generated using the ThruPLEX® DNA-seq kit (Takara bio) and sequenced by paired-end sequencing with a 75-cyce high-output Nextseq 550 kit (Illumina). All histone modification ChIP-seq was performed using exogenous reference controls using Drosophila S2 fixed chromatin based on published protocols. Briefly, fixed Drosophila S2 cells were added to fixed Ewing sarcoma cell lines at the lysis stage, prior to sonication at a ratio of 1:4 (S2:Ewing sarcoma cells).

## Animal studies

For the cell line xenograft study, $1 \times 10^6$ SKNMC cells were subcutaneously injected into NSG mice (NSG-NOD. Cg-Prkdcscid II2rgtm1Wjl/SzJ, #005557 Jackson Laboratory) with 20% Matrigel. SKNMC were negative for rodent pathogens and were confirmed mycoplasma negative. Tumor volume was calculated (volume = long diameter$^2$ × short diameter × 0.5) and measured using calipers. When 100 mm$^3$ was reached, animals were then randomized into two groups ($n = 10$ per group) and were treated once daily with vehicle only (20% Hydroxypropyl-β-cyclodextrin (HPBCD)) or 100 mg/kg A-485 by Intraperitoneal (IP) injection. Mice were treated for up to 23 days or until tumor volume reached the maximum experimental end-point size of 2 cm$^3$. No blinding was performed during this experiment. No mice lost weight or demonstrated any evidence of side effects to the drug during the experiment. Two mice per group were excluded from the analysis due to no tumor engraftment. Tumors were dissected and cells purified. Histone extraction and RNA purification were performed on at least 6 mice from each condition, excluding samples with extensive tumor necrosis and dead cell contamination.

## ChIP-seq analysis

For ChIP-seq analysis, raw sequencing reads were converted to FASTQ file format using bcl2fastq (v2.20.0.422). A summary of total unique and unique mapped reads, (along with calculated Drosophila normalization factors) are outlined in Table EV3. Reads were mapped to the human genome (Gencode GRCh38/hg38), as well as the Drosophila genome (ISO1-MT/dm6) using STAR (TDF files were made using IGVtools (v2.3.75); params –alignIntronMax 1 –alignEndsType EndToEnd – alignMatesGapMax 2000 for ChIP-seq analysis). Reads were sorted and duplicates were removed using picard pipeline tools (v2.9.4). BAM files of unique mapped reads were generated using SAMtools (v1.95).

For ChIP-seq (where there was a control and treatment), treatment BAM files were down-sampled to reflect either read normalization or, where an exogenous reference was used, the Drosophila normalization factors. Drosophila read normalization was calculated as outlined in Orlando et al (Orlando et al, 2014). All downstream analysis and interpretation were performed on normalized BAM files. See Table EV3 for total ChIP-seq reads for each sample and Drosophila normalization factors. TDF files were produced using IGVtools (TDF signal pileups; v2.3.75) for ChIP-seq track visualization using IGV. For tornado plot visualization deeptools (tornado plots; v3.1.3) was used. Tornado plots were generated at all EWS::FLI1 enhancer regions centered around the EWS::FLI1 peak with a $+/-$ 3 kb window.

EWS::FLI1 enhancer loci were defined using a previously published set of chromosome co-ordinates (Riggi et al, 2014). Enhancers were linked to the nearby gene based on proximity to the nearest promoter (Table EV2 for enhancer loci and associated genes). Enhancer co-ordinates were lifted over from hg19 to hg38 format using the UCSC Genome Browser liftover tool (https://genome.ucsc.edu/cgi-bin/hgLiftOver). Loci co-ordinates shorter than 600 bp were extended to 600 bp in order to assess histone modification and chromatin protein occupancy around the EWS::FLI1 binding site.

ChIP-seq and RNA-seq signal (for eRNA analysis) at EWS::FLI1 enhancers were calculated using bedtools (v.228.0). For all ChIP-seq analysis comparisons at loci where both control and treated sample signals were below 30rpk was excluded. Similarly, for RNA signal assessment, this threshold was reduced to 10rpk due to low signal expectation for eRNA detection. See Datasets EV4–6 for

bedtools ChIP-seq analysis at enhancer loci. Motif analysis was conducted using the Homer tool findMotifsGenome.pl using the parameters -size 200 and -mask.

## RNA-seq analysis

For RNA-seq data, raw sequencing reads were converted to FASTQ file format using bcl2fastq (v2.20.0.422). Reads were mapped to the human genome (Gencode GRCh38/hg38 using STAR (v2.7.5a). Reads were sorted and duplicates were removed using picard pipeline tools (v2.9.4). BAM files of unique mapped reads were generated using SAMtools (v1.95). Raw per-gene counts were calculated using HTSeq (htseq-count v0.6.1pl). Differential gene expression was calculated using the Bioconductor DESeq2 package (v1.24.0) from raw per-gene counts. Heatmaps of z-scores from RNA-seq data were generated using the Morpheus tool (https://software.broadinstitute.org/morpheus).

## GSEA analysis

For dot plots, Gene set enrichment analysis (GSEA) was performed using the GSEA function within the clusterprofiler (ver 4.10.1) package where parameters were set as eps = 0, pAdjustMethod = "BH" and by = "fgsea". Analysis was done with R version 4.3.2. For enrichment plots the standard GSEA (ver 4.2.3) was performed using GSEAPreranked tool using the c2.all.V2023.2.Hs.Symbols collection with 1000 permutations.

## Data availability

All sequencing data from this study are available via Gene Expression Omnibus (GEO) under accession number GSE266633 and GSE266666.

The source data of this paper are collected in the following database record: biostudies:S-SCDT-10_1038-S44319-025-00552-z.

## Peer review information

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

## Acknowledgements

We would like to thank the members of the Armstrong lab for their experimental guidance and support. We thank Dr. Alba Rodriguez-Meira for insightful feedback and reading of the manuscript. This research was funded by a grant from the NIH/NCI U54 CA231637 (SA Armstrong, K Stegmaier, and MN Rivera). MNR is supported by the Thomas F and Diana L Ryan MGH Research Scholar Award. DVW was supported by the German Research Foundation (DFG, 511811315). MB was supported by the Fonds de Recherche du Québec-Santé postdoctoral fellowship (Bf7-326051).

## Author contributions

**Laura C Godfrey**: Conceptualization; Data curation; Formal analysis; Validation; Investigation; Visualization; Methodology; Writing—original draft; Writing—review and editing. **Brandon Regalado**: Data curation; Formal analysis; Investigation. **Sydney R Schweber**: Data curation; Validation; Investigation. **Charles Hatton**: Software; Formal analysis; Visualization. **Daniela V Wenge**: Formal analysis; Investigation; Methodology. **Yanhe Wen**: Formal analysis.

**Meaghan Boileau**: Formal analysis; Visualization. **Maria Wessels**: Data curation; Formal analysis; Investigation. **Jun Qi**: Resources. **Christopher J Ott**: Resources; Writing—review and editing. **Kimberly Stegmaier**: Conceptualization; Resources; Methodology; Writing—review and editing. **Miguel N Rivera**: Conceptualization; Resources; Supervision; Writing—review and editing. **Scott A Armstrong**: Conceptualization; Resources; Supervision; Funding acquisition; Methodology; Writing—review and editing.

Source data underlying figure panels in this paper may have individual authorship assigned. Where available, figure panel/source data authorship is listed in the following database record: biostudies:S-SCDT-10_1038-S44319-025-00552-z.

## Disclosure and competing interests statement

SAA has been a consultant and/or shareholder for Neomorph Inc., C4 Therapeutics, Accent Therapeutics, Nimbus Therapeutics, AstraZeneca and Hyku Therapeutics. SAA has received research support from Janssen and Syndax. SAA is an inventor on a patent application related to MENIN inhibition WO/2017/132398A1. KS received grant funding from the DFCI/Novartis Drug Discovery Program and from KronosBio and is a member of the SAB and has stock options with Auron Therapeutics on topics unrelated to this work.

# Expanded View Figures

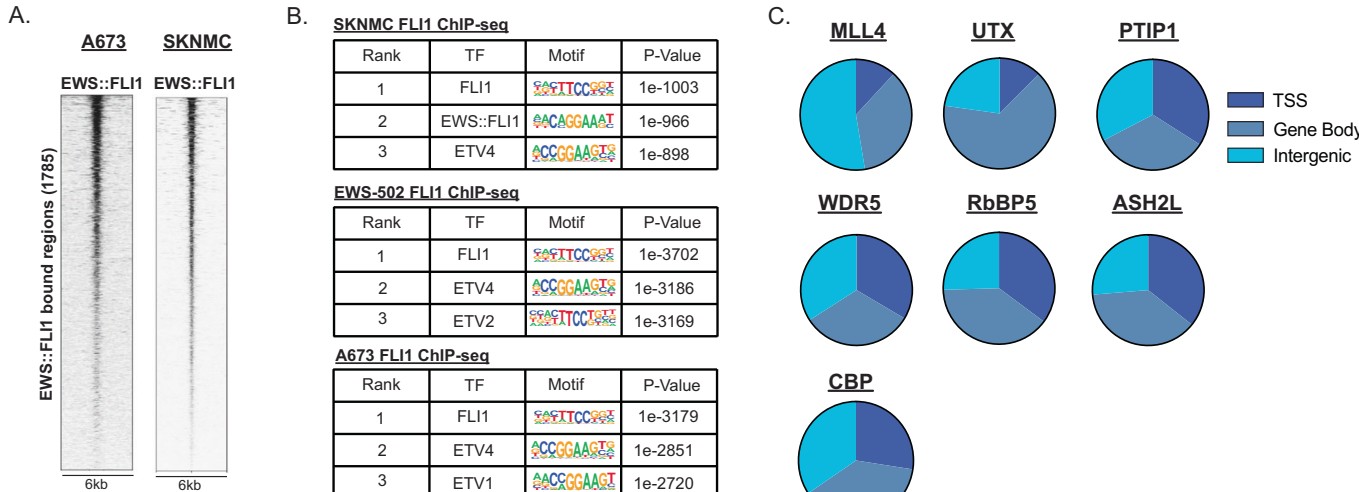

A. 

**A673**  **SKNMC**
EWS::FLI1  EWS::FLI1

EWS::FLI1 bound regions (1785)

6kb   6kb

B.

**SKNMC FLI1 ChIP-seq**

| Rank | TF | Motif | P-Value |
|------|------|-------|---------|
| 1 | FLI1 | | 1e-1003 |
| 2 | EWS::FLI1 | | 1e-966 |
| 3 | ETV4 | | 1e-898 |

**EWS-502 FLI1 ChIP-seq**

| Rank | TF | Motif | P-Value |
|------|------|-------|---------|
| 1 | FLI1 | | 1e-3702 |
| 2 | ETV4 | | 1e-3186 |
| 3 | ETV2 | | 1e-3169 |

**A673 FLI1 ChIP-seq**

| Rank | TF | Motif | P-Value |
|------|------|-------|---------|
| 1 | FLI1 | | 1e-3179 |
| 2 | ETV4 | | 1e-2851 |
| 3 | ETV1 | | 1e-2720 |

C.

MLL4   UTX   PTIP1

TSS
Gene Body
Intergenic

WDR5   RbBP5   ASH2L

CBP

D.

**A673 CBP ChIP-seq**

| Rank | TF | Motif | P-Value |
|------|------|-------|---------|
| 1 | FLI1 | | 1e-1004 |
| 2 | EWS::ERG | | 1e-990 |
| 3 | ETV2 | | 1e-989 |

**A673 p300 ChIP-seq**

| Rank | TF | Motif | P-Value |
|------|------|-------|---------|
| 1 | ETV2 | | 1e-2675 |
| 2 | FLI1 | | 1e-2630 |
| 3 | EWS::ERG | | 1e-2625 |

**A673 MLL4 ChIP-seq**

| Rank | TF | Motif | P-Value |
|------|------|-------|---------|
| 1 | FLI1 | | 1e-564 |
| 2 | ETV2 | | 1e-545 |
| 3 | EWS::FLI1 | | 1e-537 |

E.

**SKNMC**

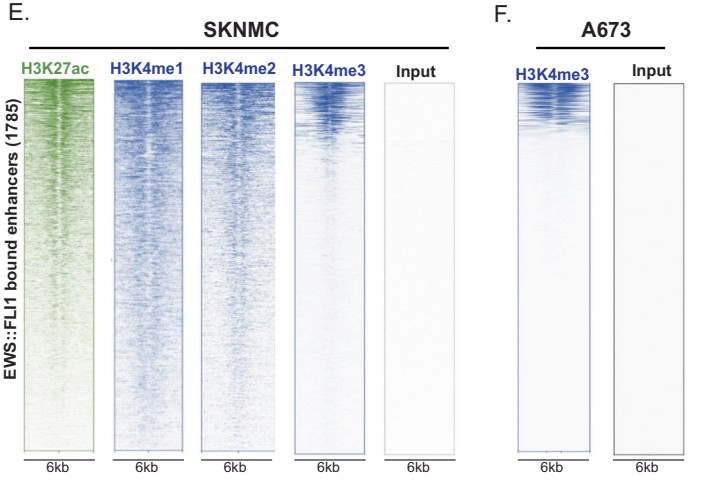

EWS::FLI1 bound enhancers (1785)

H3K27ac   H3K4me1   H3K4me2   H3K4me3   Input

6kb   6kb   6kb   6kb   6kb

F.

**A673**

H3K4me3   Input

6kb   6kb

G.

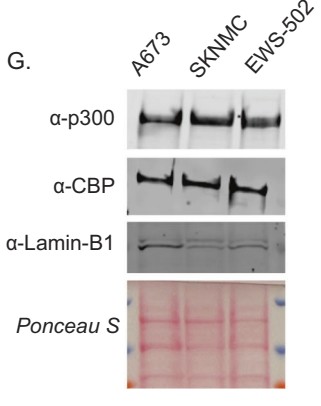

A673   SKNMC   EWS-502

α-p300

α-CBP

α-Lamin-B1

*Ponceau S*

**Figure EV1.  p300/CBP occupy EWS::FLI1 enhancer regions and mirror EWS::FLI1 binding intensity.**

(A) Tornado plots of FLI1 ChIP-seq in A673 and SKNMC cell lines at 1785 defined EWS::FLI1 enhancer regions. (B) Homer Motif analysis showing top 3 enriched motifs in FLI1 ChIP-seq in SKNMC, EWS-502 and A673 cell lines. (C) Genomic location of ChIP-seq peaks called using MACS2 of chromatin factors in A673 cells. (D) Homer Motif analysis showing top 3 enriched motifs in CBP, p300 and MLL4 ChIP-seq in A673 cells. (E) Tornado plots showing histone modification ChIP-seq signal in SKNMC cells at EWS::FLI1 enhancer regions. Signal compared to input. (F) Tornado plots for H3K4me3 ChIP-seq and associated input in A673 cells at EWS::FLI1 enhancer regions. (G) Western blot analysis of p300 and CBP in steady-state A673, SKNMC and EWS-502 cells.

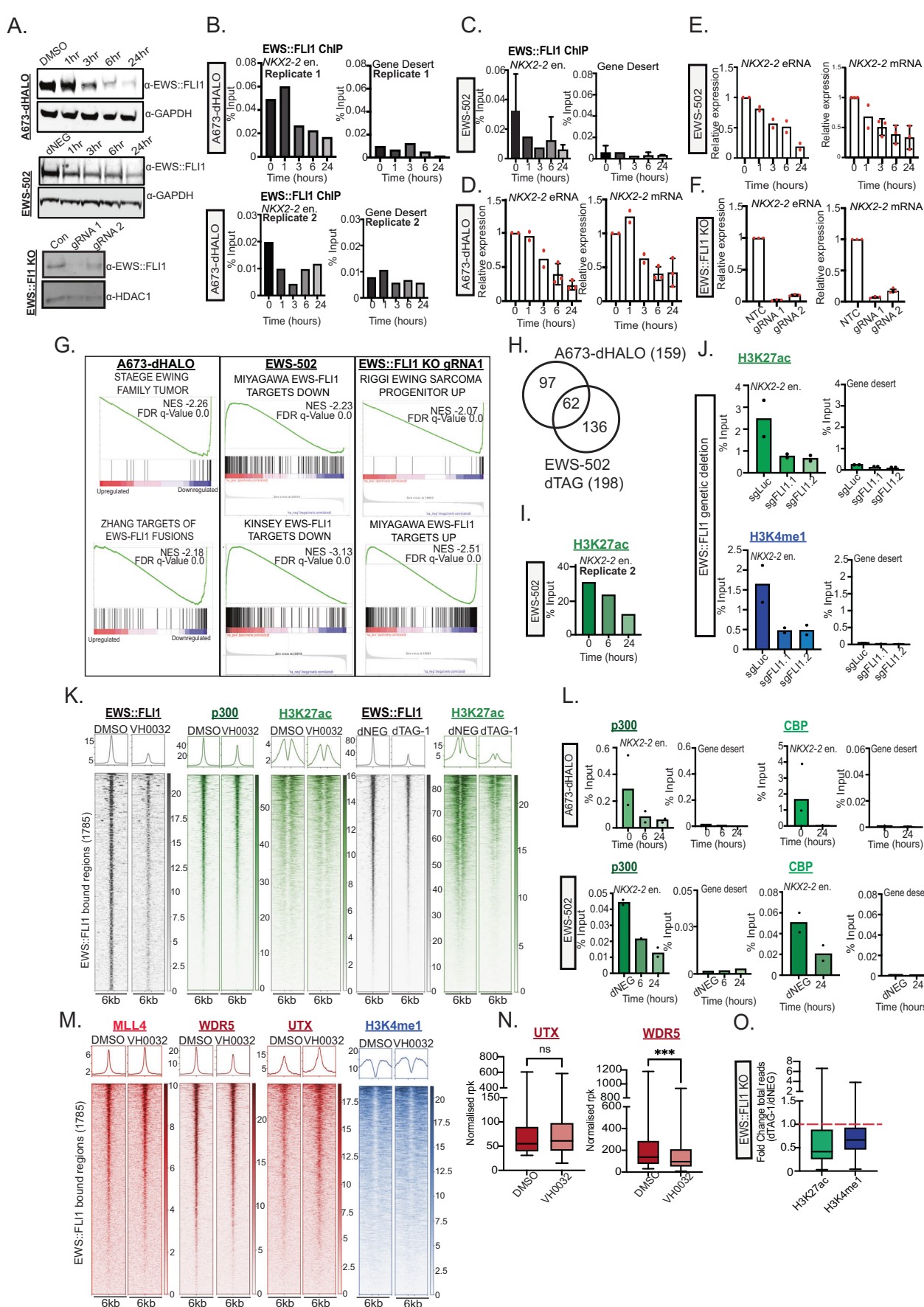

◀ **Figure EV2.  Rapid degradation of EWS::FLI1 is acutely linked to reduced p300 occupancy and H3K27ac.**

(A) FLI1 western blot analysis of in A673-dHALO cells following 1 μM VH0032 treatment, EWS-502 dTAG cells following 1 μM dTAG-1 treatment and A673 cells following FLI1 genetic deletion for 7 days. (B) FLI1 ChIP-qPCR duplicates from two independent experiments in A673-dHALO cells following 1 μM VH0032 treatment. (C) FLI1 ChIP-qPCR in EWS-502 dTAG cells following 1 μM dTAG-1 treatment from two biological replicates. (D–F) RT-qPCR in (D) A673-dHALO cells following 1 μM VH0032 treatment. (E) EWS-502 dTAG system following 1 μM dTAG-1 treatment and (F) RT-qPCR in A673 cells following 7 days EWS::FLI1 knockout using two independent gRNAs. Error bars represent standard error of the mean from three biological replicates. (G) GSEA plots showing negatively enriched EWS::FLI1 gene expression signatures in A673-dHALO VH0032 treated cells, EWS-502 dTAG-1 treated cells and EWS::FLI1 KO in A673 cells. (H) Overlap of downregulated EWS::FLI1-sensitive gene targets between A673-dHALO and EWS-502 dTAG degrader cell lines. (I) H3K27ac ChIP-qPCR replicate 2 following 6- and 24 h of EWS::FLI1 degradation in EWS-502 dTAG system (for replicate 1 see Fig. 2). (J) H3K27ac and H3K4me1 ChIP-qPCR following 7 days EWS::FLI1 genetic deletion from two independent biological replicates. (K) Tornado plots showing p300, FLI1 and H3K27ac ChIP-seq signal at EWS::FLI1 enhancers in A673-dHALO and EWS-502 dTAG systems following 24 h degradation. A673-dHALO control plots as in Fig. 1A,B. (L) p300 and CBP ChIP-qPCR after 24 h degradation in both degrader settings from at least two biological replicates. (M) MLL4, WDR5, UTX and H3K4me1 ChIP-seq signal at EWS::FLI1 enhancers after 24 h 1 μM VH0032 treatment in A673-dHALO system. A673-dHALO control plots as in Fig. 1A,B. (N) UTX and WDR5 normalized rpk at EWS::FLI1-sensitive enhancers in A673-dHALO system. Mann–Whitney *U* test, ns *P* = 0.382, ***$P$ = 0.0005. Boxes depict the range between the first and third quartile. Central line within box shows the median value and whiskers highlight the maximum to minimum data points. ChIP-seq from one biological replicate. (O) Fold change in H3K27ac and H3K4me1 ChIP-seq signal at EWS::FLI1 enhancers following EWS::FLI1 KO in A673 cells. Red line depicts steady-state levels. Boxes depict the range between the first and third quartile. Central line within box shows the median value and whiskers highlight the maximum to minimum data points. ChIP-seq from one biological replicate.

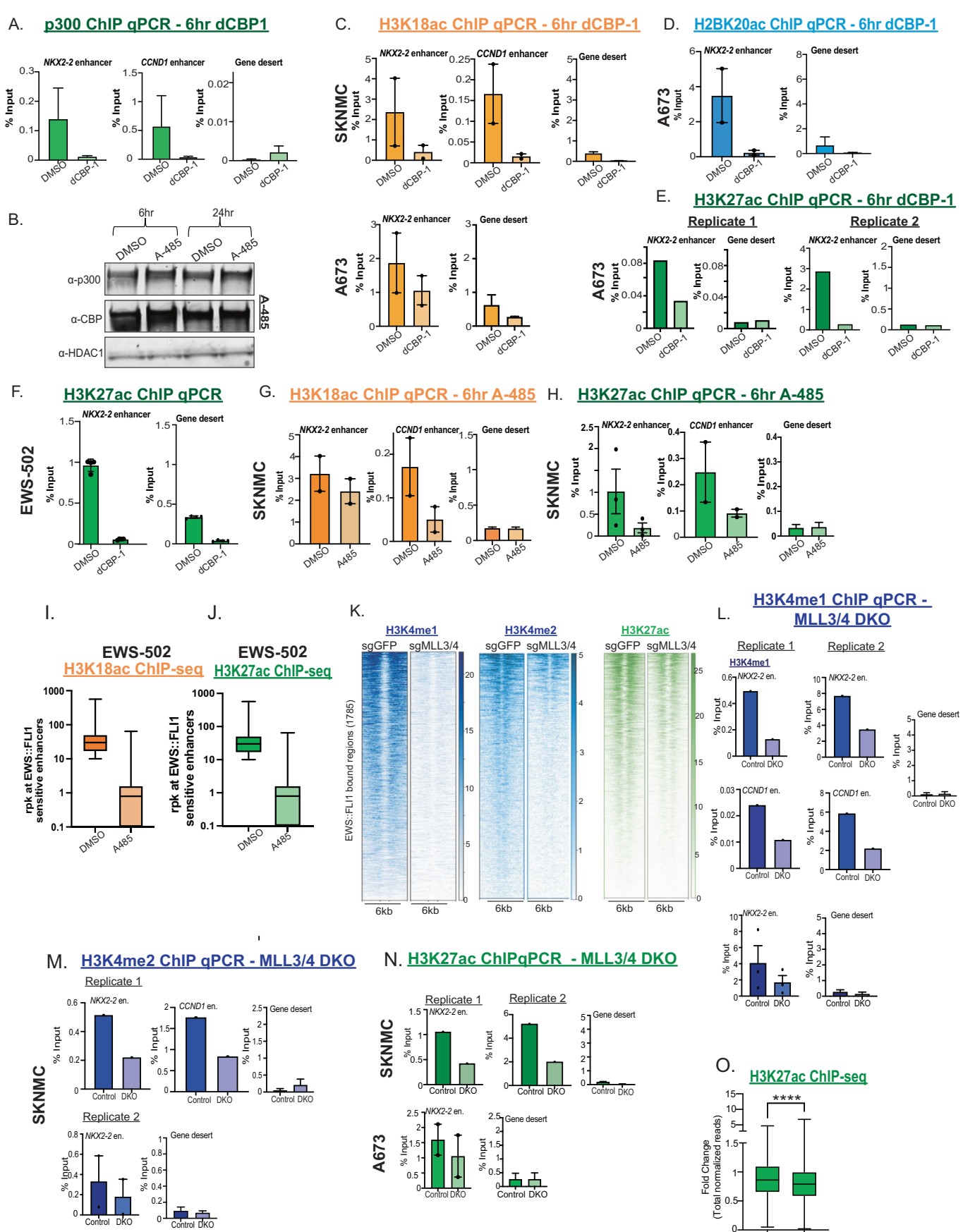

**Figure EV3.  Loss of p300/CBP and MLL3/4 protein function alter the EWS::FLI1 enhancer histone modification landscape.**

(A) p300 ChIP-qPCR in DMSO control and dCBP-1 conditions in SKNMC cells from two biological replicates. (B) p300 and CBP western blot from nuclear extracted SKNMC cells treated with A-485 for 6 h and 24 h. (C) H3K18ac ChIP-qPCR in SKNMC and A673 following 6 h dCBP-1 treatment from two biological replicates. (D, E) H2BK20ac and H3K27ac ChIP-qPCR at NKX2-2 enhancer in SKNMC and A673 following 6 h dCBP-1 treatment. Graphs shown for two biological replicates. (F) H3K27ac ChIP-qPCR at NKX2-2 enhancer following dCBP-1 (6 h) treatment from at least three biological replicates. (G, H) H3K18ac and H3K27ac ChIP-qPCR at *NKX2-2* and *CCND1* enhancer in SKNMC cells following 6 h A-485 treatment from at least two biological replicates. Error bars where present depict standard error of the mean from three biological replicates. (I) H3K18ac normalized rpk at EWS::FLI1 enhancers in DMSO and A-485 (6 h) condition in EWS-502 cells. ChIP-seq from one biological replicate. (J) H3K27ac normalized rpk at EWS::FLI1 enhancers in DMSO and A-485 (6 h) condition in EWS-502 cells. ChIP-seq from one biological replicate. (K) Histone modifications at EWS::FLI1 enhancers in sgGFP control and sgMLL3/4 DKO conditions in A673 cells. (L) H3K4me1 ChIP-qPCR in sgGFP control and sgMLL3/4 DKO conditions in SKNMC and A673 cells from two biological replicates. (M) H3K4me2 ChIP-qPCR in sgGFP control and sgMLL3/4 DKO conditions in SKNMC cells from two biological replicates. (N) H3K27ac ChIP-qPCR in sgGFP control and sgMLL3/4 DKO conditions in SKNMC and A673 cells from two biological replicates. (O) H3K27ac ChIP-seq fold change at all EWS::FLI1 enhancers in MLL3/4 DKO and A-485 (6 h) treatment compared to DMSO control in SKNMC cells. Kruskal–Wallis test, ****$P < 0.0001$. Boxes depict the range between the first and third quartile. Central line within box shows the median value and whiskers highlight the maximum to minimum data points. ChIP-seq from one biological replicate.

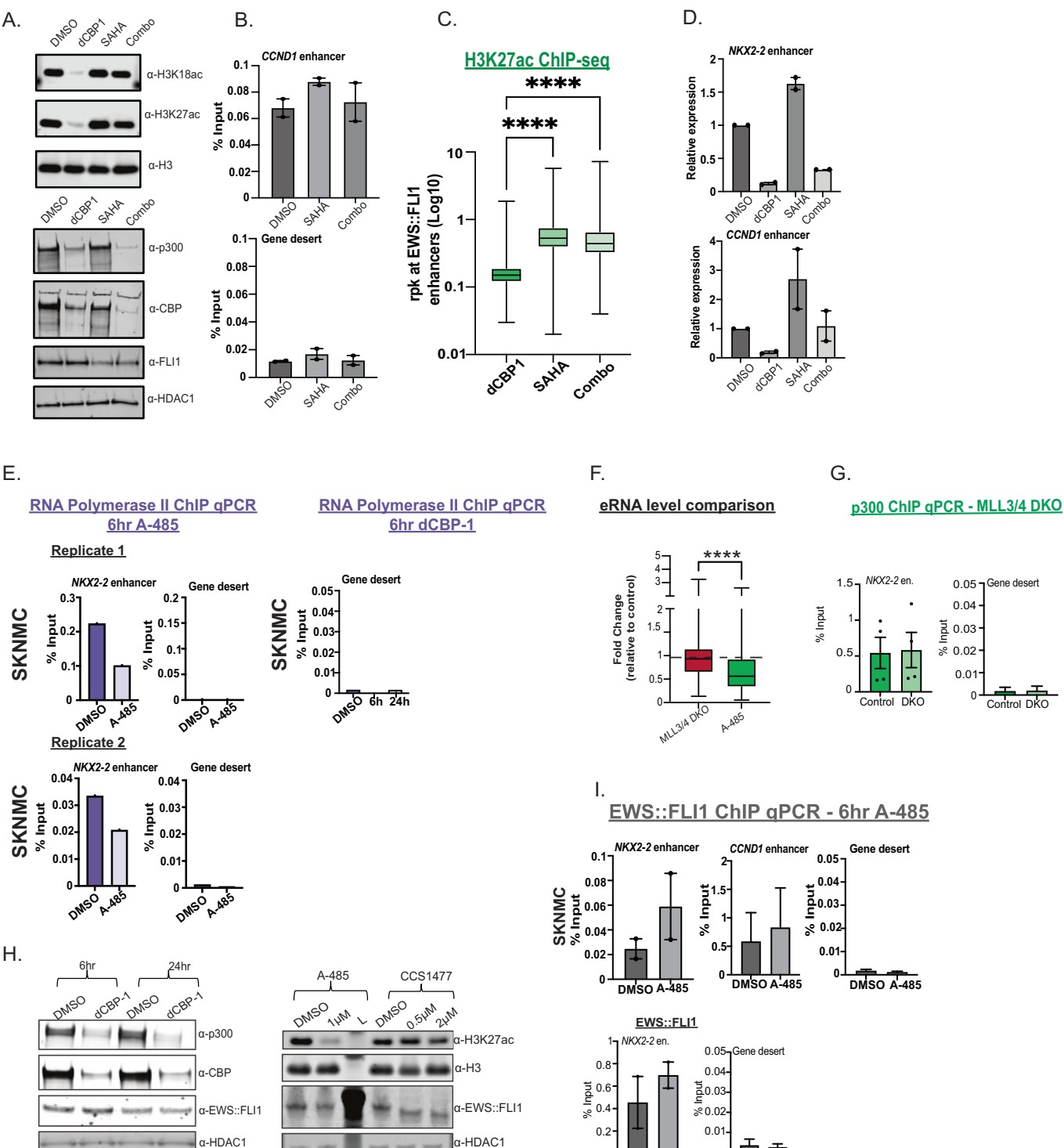

Figure EV4. p300/CBP is required for enhancer RNA levels and RNA polymerase II occupancy at EWS::FLI1 enhancers independently of EWS::FLI1 binding.

(A) Western blot analysis in dCBP-1, SAHA and combined SAHA and dCBP-1 treatment. (B) FLI1 ChIP-qPCR in DMSO, SAHA and combined SAHA and dCBP-1 treatment. Graphs depict two biological replicates. (C) H3K27ac ChIP-seq normalized rpk at EWS::FLI1 enhancers in dCBP-1, SAHA and combined SAHA and dCBP-1 treatment. Boxes depict the range between the first and third quartile. Central line within box shows the median value and whiskers highlight the maximum to minimum data points. ChIP-seq from one biological replicate. (D) RT-qPCR at EWS::FLI1 enhancers in SKNMC cells following 24 h treatment of either 100 nM dCBP-1, 1 μM SAHA or dual treatment from two biological replicates. (E) RNA polymerase II ChIP-qPCR at $NKX2-2$ enhancer in SKNMC cells following 6 h A-485 and 6 h dCBP-1 treatment from two biological replicates. (F) Fold change in eRNA RNA-seq reads at EWS::FLI1-sensitive enhancers following MLL3/4 DKO compared to A-485 treatment. Mann–Whitney $U$ test, ****$P < 0.0001$. Boxes depict the range between the first and third quartile. Central line within box shows the median value and whiskers highlight the maximum to minimum data points. eRNA expression calculated from three biological replicates. (G) p300 ChIP-qPCR in sgGFP control and sgMLL3/4 DKO conditions in A673 cells. Error bars represent standard error of the mean from at least three biological replicates. (H) Western blot analysis of EWS::FLI1 from nuclear extracts of SKNMC cells treated with dCBP-1, A-485 or CCS1477. 'L' indicates protein ladder lane. (I) EWS::FLI1 ChIP-qPCR at $NKX2-2$ and $CCND1$ enhancers in SKNMC cells following 6 h A-485 treatment from two biological replicates.

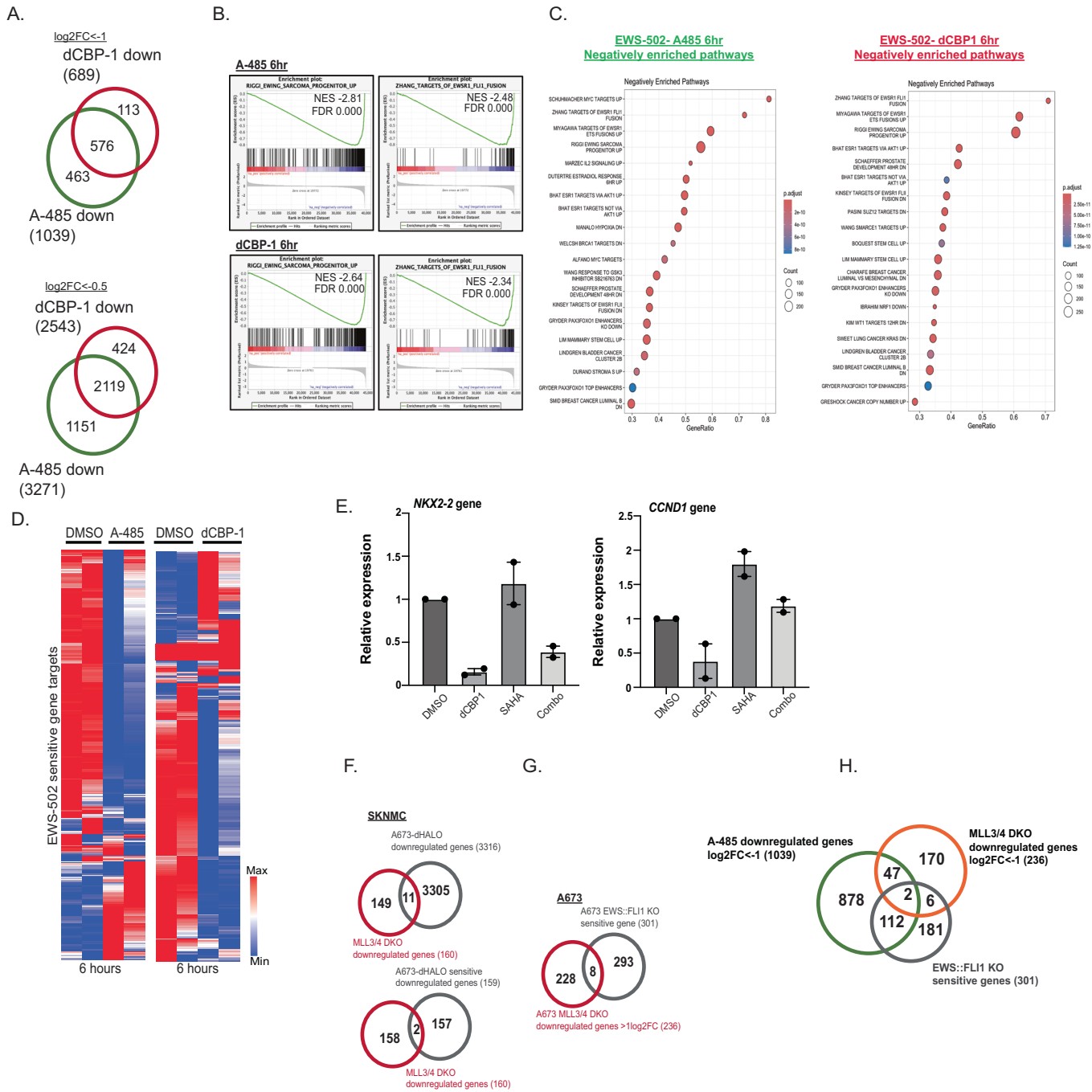

**Figure EV5. p300/CBP function are central to the EWS::FLI1 gene regulatory network.**

(A) Overlap between downregulated genes in A-485 and dCBP-1 conditions at two different log2FC cut off thresholds. (B) GSEA negatively enriched EWS::FLI1 signatures in both conditions. (C) Gene set enrichment scores showing top 20 negatively enriched gene sets following 6 h A-485 and 6 h dCBP-1 treatment, in order of enrichment in EWS-502 cells. Color represents *P*.adjust values from a Benjamini-Hochberg (BH) procedure test. Gene Ratio represents total differentially expressed genes in given gene set. (D) Z-scores of EWS::FLI1-sensitive enhancers and genes in DMSO, dCBP-1 and A-485 treatment conditions. (E) RT-qPCR at EWS::FLI1 genes in SKNMC cells following 24 h treatment of either 100 nM dCBP-1, 1 μM SAHA or dual treatment from two biological replicates. (F) Overlap between MLL3/4 DKO downregulated genes in SKNMC cells with genes downregulated in A673-dHALO setting and EWS::FLI1-sensitive genes. (G) Overlap between MLL3/4 DKO downregulated genes in SKNMC cells with genes downregulated in EWS::FLI1 KO setting in A673 cells. (H) Overlap between A-485 downregulated genes (either log2FC < -1 or log2FC < −0.5), MLL3/4 DKO downregulated genes (either log2FC < −1 or log2FC < -0.5) and EWS::FLI1 KO sensitive genes.

A.

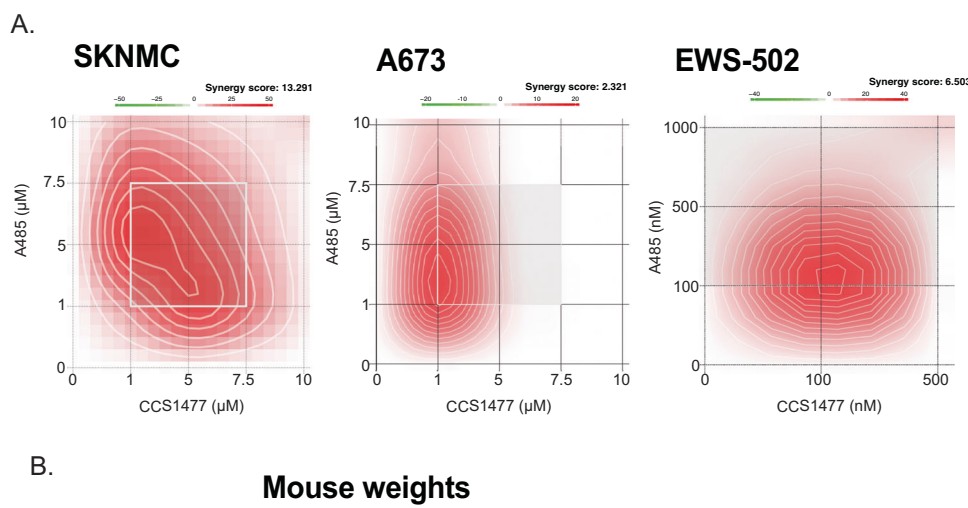

B.

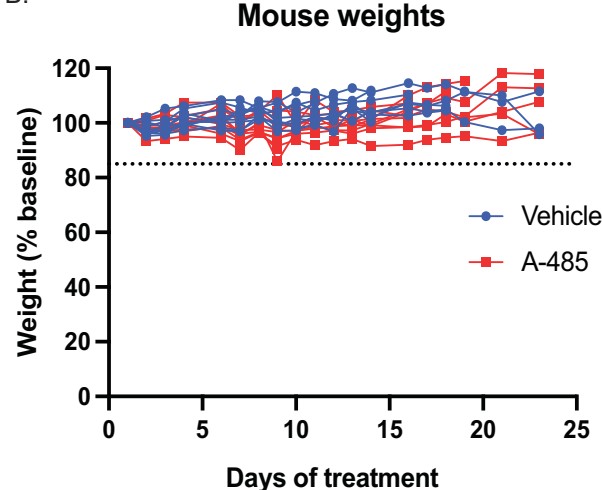

Figure EV6.  p300/CBP inhibition decelerates Ewing sarcoma tumor growth in vivo.

(A) Synergy plots for combination treatment of A-485 and CCS1477 in A673, SKNMC and EWS-502 cells. (B) Percentage weight of mice compared to day 1 of treatment in vehicle-only (blue) and 100 mg/kg A-485 (red) cohorts. Each point and line represent a single mouse.

