## [Peer Review File · EMBO Reports]

p300/CBP is an essential driver of pathogenic enhancer activity and gene expression in Ewing sarcoma

Laura Godfrey, Brandon Regalado, Sydney Schweber, Charles Hatton, Daniela Wenge, Yanhe Wen, Meaghan Boileau, Maria Wessels, Jun Qi, Christopher Ott, Kimberly Stegmaier, Miguel Rivera, and Scott Armstrong

Corresponding author: Scott Armstrong (scott_armstrong@dfci.harvard.edu)

Review Timeline:

Submission Date:	4th Oct 24
Editorial Decision:	13th Nov 24
Revision Received:	3rd Jun 25
Editorial Decision:	8th Jul 25
Revision Received:	11th Jul 25
Accepted:	4th Aug 25

Editor: Esther Schnapp

Transaction Report:

Dear Dr. Armstrong,

Thank you for the submission of your manuscript to EMBO reports. We have now received the full set of referee reports that is pasted below.

As you will see, the referees acknowledge that the findings are potentially interesting and they acknowledge the technical quality of your work. However, they have also several suggestions for how the study should be strengthened and I think all suggestions are good and should be addressed. Please let me know in case you disagree, and we can discuss the exact revision requirements further, also in a video chat, if you like.

I would thus like to invite you to revise your manuscript with the understanding that the referee concerns must be fully addressed and their suggestions taken on board. Please address all referee concerns in a complete point-by-point response. Acceptance of the manuscript will depend on a positive outcome of a second round of review. It is EMBO reports policy to allow a single round of major revision only and acceptance or rejection of the manuscript will therefore depend on the completeness of your responses included in the next, final version of the manuscript.

We realize that it is difficult to revise to a specific deadline. In the interest of protecting the conceptual advance provided by the work, we recommend a revision within 3 months (13th Feb 2025). Please discuss the revision progress ahead of this time with the editor if you require more time to complete the revisions.

- 1) A data availability section providing access to data deposited in public databases is missing. If you have not deposited any data, please add a sentence to the data availability section that explains that.
- 2) Your manuscript contains statistics and error bars based on $n=2$. Please use scatter blots in these cases. No statistics should be calculated if $n=2$.

3) We replaced Supplementary Information with Expanded View (EV) Figures and Tables that are collapsible/expandable online. A maximum of 5 EV Figures can be typeset. EV Figures should be cited as 'Figure EV1, Figure EV2' etc... in the text and their respective legends should be included in the main text after the legends of regular figures.

5) a complete author checklist, which you can download from our author guidelines <https://www.embopress.org/page/journal/14693178/authorguide>. Please insert information in the checklist that is also reflected in the manuscript. The completed author checklist will also be part of the RPF.

6) Please note that all corresponding authors are required to supply an ORCID ID for their name upon submission of a revised manuscript (<https://orcid.org/>). Please find instructions on how to link your ORCID ID to your account in our manuscript tracking system in our Author guidelines <https://www.embopress.org/page/journal/14693178/authorguide#authorshipguidelines>

10) Regarding data quantification (see Figure Legends:

<https://www.embopress.org/page/journal/14693178/authorguide#figureformat>)

12) All Materials and Methods need to be described in the main text using our 'Structured Methods' format, which is required for all research articles. According to this format, the Methods section includes a separate file Reagents and Tools Table (listing key reagents, experimental models, software and relevant equipment and including their sources and relevant identifiers) and a Methods and Protocols section describing the methods using a step-by-step protocol format. The aim is to facilitate adoption of the methodologies across labs. More information on how to adhere to this format as well as a downloadable template (.docx) for the Reagents and Tools Table can be found in our author guidelines:

An example of a Method paper with Structured Methods can be found here: <https://www.embopress.org/doi/full/10.1038/s44320-024-00037-6#sec-4>

I look forward to seeing a revised form of your manuscript when it is ready.

Yours sincerely,

Referee #1:

In this study, Godfrey et al. studied the role of p300/CBP and of the MLL3/4 complex in EWS::FLI1 driven gene regulation. They performed ChIP-seq, eRNA and mRNA expression studies, both genome-wide and focused on selected EWS::FLI1 driven enhancers. They employed A673 and SK-N-MC Ewing sarcoma cell lines with MLL3/4 knock-out at 7 days post gene editing and two previously published genetically engineered Ewing sarcoma cell line models with degradable EWS::FLI1 protein expression at 3, 6 and 24 hours of EWS::FLI1 depletion. They provide evidence for p300/CBP but not MLL3/4 to critically impact EWS::FLI1-driven enhancer activity without affecting EWS::FLI1 binding stability. These data were validated using three p300 targeting compounds, a degrader, an inhibitor of the enzymatic activity and an inhibitor of the bromodomain. Finally, they demonstrate in vitro and in a xenograft mouse model that p300/CBP enzymatic inhibition using A-485 led to significant deceleration of tumor growth identifying p300/CBP as a potential therapeutic target in Ewing sarcoma.

This study provides mechanistic insights and independent confirmation of a recently published (DOI: 10.1186/s12943-024-02115-7) essential role for p300/CBP in EWS::FLI1 mediated gene activation. It is thoroughly performed and provides the basis for further pre-clinical development of p300/CBP targeting compounds for the treatment of Ewing sarcoma.
Major:

The experiment of inhibition of genome-wide deacetylation by SAHA combined with p300/CBP degradation shown in Fig. EV 4A and Fig. EV 5D is not conclusive. The interpretation that partial rescue of EWS::FLI1 dependent eRNA expression is mediated via a p300/CBP activity independent of its acetylation function is pre-mature as it is possible that the p300/CBP degrader reduced target levels much faster than SAHA-mediated inhibition of deacetylation. One would need to pretreat the cells with SAHA before adding the p300/CBP degrader to exclude a decrease in acetylation before complete inhibition of deacetylases. Also, ChIP-seq should be performed to exclude decreased acetylation at EWS::FLI1 target enhancers in the double-treated versus SAHA-alone treated cells. Finally, EWS::FLI1 levels should be monitored in all compound treated cells (including SAHA) to exclude alterations in its expression.

Minor:

- i) Fig. 4H: How do the authors explain the increase in EWS::FLI1 occupancy caused by MLL3/4 double knock-out?
- ii) Fig 4I: Not clear, which conditions individual tracks reflect. If it is MLL3/4 DKO, why is there no change in H3K4me1?
- iii) Numerical in-text references in the first paragraph on page 4 cannot be assigned to alphabetical reference list.
- iv) Reference list is incomplete (i.e. lacks references from Materials and Methods).
- v) Fig EV2B shows only a single replicate for quantification of EWS::FLI1 ChIP-PCR on NKX2-2 enhancer.
- vi) Text refers to Fig 2C-D for eRNA expression after 3 hours of EWS::FLI1 degradation in the A673-dHALO system. However, this figure demonstrates the effect of 24h VH0032 treatment on NKX2-2 and VRK1 loci.
- vii) Figure legend to Fig 2E: refers to Fig EV 2I instead of Fig EV 2H for second replicate of cell line EWS-502.

Referee #2:

This manuscript by Godfrey et al., describes a detailed analysis of chromatin modifying complexes in enhancer regulation in the Ewing sarcoma tumor. They specifically address the contributions of the MLL3/4 and p300/CBP complexes to enhancer activation and function in regulation of EWS::FLI1 target genes.

The work is robust, rigorous, and compelling. Multiple orthogonal approaches are used, including genetically modified and pharmacologically inhibited cells. Despite an overwhelming amount of data, the manuscript is very clear, well organized and concise, and easy to read. The conclusions are justified, and the findings provide new and important insights into enhancer regulation in Ewing sarcoma.

A few minor corrections are suggested to improve clarity:

- 1) Please move the spaghetti plots of tumor growth (EV6B) to the main figure (replace current 6C if needed to allow space)
- 2) Please give specific rationale for evaluating the different acetylation marks (ie the p300/CBP acetylation marks H3K18Ac and H2ABK120 are not as well described or known by most investigators, and it would help readers to understand why these marks were specifically selected)
- 3) There are so many panels in the figures and so many different marks are evaluated in different cell lines - it would help with

clarity if the authors could label each of the panels rather than only using figure legends.

Referee #3:

Summary: In the manuscript by Armstrong and colleagues, the authors seek to understand the mechanism through which the oncogenic fusion protein, EWS::FLI1, activates pathogenic enhancers. The authors examined whether EWS::FLI1 interacts with MLL3/4 methyltransferases and/or CBP/P300 acetyl transferases to mediate enhancer activation using RT-qPCR, ChIP-seq, as well as additional methods. The authors found that there are a subset of enhancers that are activated by EWS::FLI1 through CBP/P300, but not by MLL3/4. The authors furthered their findings by demonstrating that treatment of EWS cell lines in vitro with A-485 and CCS1477, two CBP/P300 inhibitors that target the enzymatic and bromodomain functions of the proteins respectively, decreased cell proliferation, and went on to show that A-485 decreased ES mouse xenograft growth in vivo.

Overall, the manuscript is rigorous and provides good evidence that EWS::FLI1 activates enhancers in large part through CBP/P300. The following issues could be addressed to strengthen the manuscript:

Figure 1:

- Motif analysis would be helpful to check that EWS::FLI1 is binding at the expected GGAA repetitive motifs, and to explore that possibility that CBP, P300, MLL3, and MLL4 are being directed to sites by other TFs.
- Are CBP and P300 both expressed in EWS to the same extent? Does it depend on the cell line?

Figure 2:

- In the two degron systems (A673 and EWS-502)- how much overlap is there between the genes regulated by EWS::FLI1?
- Where is P300 normalized rpk in figure 2H and similarly where is MLL4 in figure 2J?

Figure 3:

- Labeling could be improved in numerous places. For example, in figure 3G what are the tornado plots for? In the text it says histone acetylation, and in the figure legend it says H3K18ac- but it would help to have this labeled on the figure.
- Also in this figure, specific enhancers were referred to as dHalo sensitive, which presumably refers to EWS::FLI1 sensitive. It may be best to label that way- as it is a bit confusing when called dHalo sensitive.
- The authors should better explain why they examine the H3K18ac mark instead of H3K27ac?
- Also, since the authors wanted to compare MLL3/4 activity to CBP-P300, why did they switch to using A673 and SKNMC (rather than A673 and EWS-502)?

Figure 4:

- To strengthen the claim about the scaffolding role of CBP/P300 they get from using CCS1477, it would be helpful to see KD/genetic addback experiments, with specific domains of the protein deleted.
- CBP and P300 are referred to as largely interchangeable. There are differences between the two proteins and the authors should explain why they focusing mostly on P300 and not CBP.

Figure 6:

- In figure 6, it would be helpful to run the models to determine if A-485, CCS1477, and combo treatment are additive, synergistic, or antagonistic.

Minor issues to address:

1. In some parts of the paper, references are numbered rather than listed as author names etc.

Referee #1:

In this study, Godfrey et al. studied the role of p300/CBP and of the MLL3/4 complex in EWS::FLI1 driven gene regulation. They performed ChIP-seq, eRNA and mRNA expression studies, both genome-wide and focused on selected EWS::FLI1 driven enhancers. They employed A673 and SK-N-MC Ewing sarcoma cell lines with MLL3/4 knock-out at 7 days post gene editing and two previously published genetically engineered Ewing sarcoma cell line models with degradable EWS::FLI1 protein expression at 3, 6 and 24 hours of EWS::FLI1 depletion. They provide evidence for p300/CBP but not MLL3/4 to critically impact EWS::FLI1-driven enhancer activity without affecting EWS::FLI1 binding stability. These data were validated using three p300 targeting compounds, a degrader, an inhibitor of the enzymatic activity and an inhibitor of the bromodomain. Finally, they demonstrate in vitro and in a xenograft mouse model that p300/CBP enzymatic inhibition using A-485 led to significant deceleration of tumor growth identifying p300/CBP as a potential therapeutic target in Ewing sarcoma.

This study provides mechanistic insights and independent confirmation of a recently published (DOI: 10.1186/s12943-024-02115-7) essential role for p300/CBP in EWS::FLI1 mediated gene activation. It is thoroughly performed and provides the basis for further pre-clinical development of p300/CBP targeting compounds for the treatment of Ewing sarcoma.

Major:

The experiment of inhibition of genome-wide deacetylation by SAHA combined with p300/CBP degradation shown in Fig. EV 4A and Fig. EV 5D is not conclusive. The interpretation that partial rescue of EWS::FLI1 dependent eRNA expression is mediated via a p300/CBP activity independent of its acetylation function is premature as it is possible that the p300/CBP degrader reduced target levels much faster than SAHA-mediated inhibition of deacetylation. One would need to pretreat the cells with SAHA before adding the p300/CBP degrader to exclude a decrease in acetylation before complete inhibition of deacetylases.

We thank the reviewer for this comment and agree that further clarification is needed for this set of experiments. As suggested, we pre-treated SKNMC cells with 1 μ M SAHA for 24 hours, followed by 6 hours of 100 nM dCBP1 treatment to ensure preservation of histone acetylation.

Western blot analysis showed that bulk histone acetylation modifications (H3K27ac and H3K18ac) was slightly elevated after SAHA treatment (Figure EV4A), reduced with dCBP1 (as seen in Figure 3A), and maintained at DMSO levels in the combined SAHA + dCBP1 treatment (Figure EV4A). This suggests that SAHA pretreatment preserves histone acetylation levels, and that the dual treatment of SAHA and dCBP1 only perturbs p300 and CBP protein levels and not histone acetylation.

We also updated the RT-qPCR data at both the EWS::FLI1 target enhancers and genes in both Figure EV4D and EV5E to reflect the experimental change (24 hours pretreat of SAHA followed by 6 hours dCBP1 treatment) and can confirm that the result obtained is similar with dual treatment of SAHA and dCBP1 leading to a partial rescue of gene expression compared to dCBP1 treatment alone.

Also, ChIP-seq should be performed to exclude decreased acetylation at EWS::FLI1 target enhancers in the double-treated versus SAHA-alone treated cells.

We performed a H3K27ac ChIP-seq experiment in all conditions (DMSO, dCBP1, SAHA and combination) to assess the histone acetylation changes at the chromatin level. We observed a significant decrease in H3K27ac in dCBP1 treatment compared to the SAHA and combination treatment (Figure EV4C) demonstrating that histone acetylation is largely intact in both the SAHA and combination treatments.

Finally, EWS::FLI1 levels should be monitored in all compound treated cells (including SAHA) to exclude alterations in its expression.

EWS::FLI1 western blot show a minor decrease in EWS::FLI1 protein levels in SAHA and combination treatment (Figure EV4A). Despite this, we do not observe a decrease in gene expression levels in the SAHA treatment alone suggesting that this small decrease in FLI1 protein does not impact gene expression (Figure EV4C). To confirm this more rigorously we performed EWS::FLI1 ChIP qPCR to demonstrate that EWS::FLI1 binding is not perturbed at the CCND1 enhancer region (Figure EV4B).

Minor:

i) Fig. 4H: How do the authors explain the increase in EWS::FLI1 occupancy caused by MLL3/4 double knock-out?

We appreciate the reviewer's observation. While the reason for the consistent increase in EWS::FLI1 binding following MLL3/4 DKO remains unclear, one possible explanation is that the loss of MLL3 and MLL4, both large proteins, may increase chromatin accessibility, thereby allowing EWS::FLI1 to bind with higher affinity to target loci. A similar effect has been reported in the literature, where depletion of activating transcription factors such as CTCF resulted in increased chromatin accessibility (Xu et al., 2021).

We have now added this speculative comment to the discussion:

"Interestingly, we observed a slight increase in EWS::FLI1 binding following MLL3/4 DKO (Figure 4H-I). Previously studies have demonstrated that loss of activating transcription factors has led to sites of increased chromatin accessibility (Xu et al., 2021, Friman et al., 2019). Something similar could be occurring at EWS::FLI1 enhancers upon MLL3/4 DKO which may permit an increase in EWS::FLI1 binding affinity to DNA."

ii) Fig 4I: Not clear, which conditions individual tracks reflect. If it is MLL3/4 DKO, why is there no change in H3K4me1?

The H3K4me1 tracks in Figure 4I represent steady-state data. These tracks were inadvertently repeated in the figure, so we have removed the redundant tracks and labelled the remaining one as "steady-state" to avoid confusion.

iii) Numerical in-text references in the first paragraph on page 4 cannot be assigned to alphabetical reference list.

We have corrected this to include the appropriate references in the correct format.

iv) Reference list is incomplete (i.e. lacks references from Materials and Methods).

We have made sure all references are now included in the reference list.

v) Fig EV2B shows only a single replicate for quantification of EWS::FLI1 ChIP-PCR on NKX2-2 enhancer.

The ChIP-qPCR replicates are displayed separately for this figure due to differences in % input between the two replicates (even though the trend is the same). We have added labels to clarify this for the reader.

vi) Text refers to Fig 2C-D for eRNA expression after 3 hours of EWS::FLI1 degradation in the A673-dHALO system. However, this figure demonstrates the effect of 24h VH0032 treatment on NKX2-2 and VRK1 loci.

The RNA-seq tracks at *NKX2-2* and the *VRK1* are displayed at 3, 6 and 24hr post EWS::FLI1 degradation (Figure 2C-D). The largest effect is visible at the later 24h timepoint but there is a visible decrease in eRNA level at 3hr (bottom tracks labelled in grey).

vii) Figure legend to Fig 2E: refers to Fig EV 2I instead of Fig EV 2H for second replicate of cell line EWS-502.

With the new additional figures this subfigure is now EV2I and this has been reflected in both the text and figure legend.

Referee #2:

This manuscript by Godfrey et al., describes a detailed analysis of chromatin modifying complexes in enhancer regulation in the Ewing sarcoma tumor. They specifically address the contributions of the MLL3/4 and p330/CBP complexes to enhancer activation and function in regulation of EWS::FLI1 target genes. The work is robust, rigorous, and compelling. Multiple orthogonal approaches are used, including genetically modified and pharmacologically inhibited cells. Despite an overwhelming amount of data, the manuscript is very clear, well organized and concise, and easy to read. The conclusions are justified, and the findings provide new and important insights into enhancer regulation in Ewing sarcoma. A few minor corrections are suggested to improve clarity:

1) Please move the spaghetti plots of tumor growth (EV6B) to the main figure (replace current 6C if needed to allow space)

We have now moved this figure into the main Figure 6 and it is now Figure 6D.

2) Please give specific rationale for evaluating the different acetylation marks (ie the p300/CBP acetylation marks H3K18Ac and H2ABK120 are not as well described or

known by most investigators, and it would help readers to understand why these marks were specifically selected).

While H3K27ac has been majorly associated with p300/CBP catalytic activity, other modifications including H3K18ac and H2B acetylation have recently been highlighted as specific products of p300/CBP enzymatic activity linked to active enhancers. Furthermore, these modifications were shown to outperform H3K27ac in terms of predicting p300/CBP target genes. For these reasons we decided to assess a range of histone acetylation modifications, including H3K18ac and H2BK120ac (as an example of H2B acetylation) at EWS::FLI1 enhancers. This is now addressed in the main text to provide clarity.

“We first examined how histone acetylation is affected at EWS::FLI1 enhancers following p300/CBP perturbation. Traditionally, p300/CBP has been associated with H3K27ac and enhancer activity (Creyghton et al., 2010). However, recent studies have shown that additional acetylation marks, including H3K18ac and acetylated H2B residues, are dynamically linked to p300/CBP function and active enhancer function (Weinert et al., 2018, Narita et al., 2023). Therefore, we expanded our analysis to include H3K18ac and H2BK120ac at EWS::FLI1 enhancers”

3) There are so many panels in the figures and so many different marks are evaluated in different cell lines - it would help with clarity if the authors could label each of the panels rather than only using figure legends.

We have carefully gone through each figure to label what each subfigure is where appropriate.

Referee #3:

Summary: In the manuscript by Armstrong and colleagues, the authors seek to understand the mechanism through which the oncogenic fusion protein, EWS::FLI1, activates pathogenic enhancers. The authors examined whether EWS::FLI1 interacts with MLL3/4 methyltransferases and/or CBP/P300 acetyl transferases to mediate enhancer activation using RT-qPCR, ChIP-seq, as well as additional methods. The authors found that there are a subset of enhancers that are activated by EWS::FLI1 through CBP/P300, but not by MLL3/4. The authors furthered their findings by demonstrating that treatment of EWS cell lines in vitro with A-485 and CCS1477, two CBP/P300 inhibitors that target the enzymatic and bromodomain functions of the proteins respectively, decreased cell proliferation, and went on to show that A-485 decreased ES mouse xenograft growth in vivo.

Overall, the manuscript is rigorous and provides good evidence that EWS::FLI1 activates enhancers in large part through CBP/P300. The following issues could be addressed to strengthen the manuscript:

Figure 1:

- Motif analysis would be helpful to check that EWS::FLI1 is binding at the expected GGAA repetitive motifs, and to explore that possibility that CBP, P300, MLL3, and MLL4 are being directed to sites by other TFs.

We thank the reviewer for this comment. We performed motif analysis using the HOMER known motif algorithm. FLI1 ChIP-seq in all cell lines tested (A673, SKNMC and EWS-502) confirmed that FLI1 motifs were the most enriched motifs. In general, ETS factor motifs are similar and therefore other ETS transcription factor motifs were observed at FLI1 binding sites. For other factors including p300, CBP and MLL4, FLI1 binding sites were present within the top three enriched motifs demonstrating that these factors are also associated with FLI1 and ETS transcription factor binding. Future studies investigating the role of MLL3/4 at non EWS::FLI1 binding sites (other ETS factor sites) could prove insightful.

Are CBP and P300 both expressed in EWS to the same extent? Does it depend on the cell line?

We extracted nuclear proteins from A673, SKNMC, and EWS-502 cells and performed Western blotting for p300 and CBP. Both proteins were highly expressed across all cell lines used in this study (Figure EV1G).

Figure 2:

- In the two degron systems (A673 and EWS-502)- how much overlap is there between the genes regulated by EWS::FLI1?

We identified 62 overlapping sensitive targets in both the A673 and EWS-502 degron systems (Figure EV2H), including well-established EWS::FLI1 targets such as NKX2-2 and VRK1. Notably, differences between these degron systems may arise from variations in EWS::FLI1 degradation dynamics and cell line-specific target genes.

- Where is P300 normalized rpk in figure 2H and similarly where is MLL4 in figure 2J?

We have now performed p300 and MLL4 ChIP-seq in the EWS-502 degron system following 24 hours of dTAG-1 treatment. We have included box plots in Figure 2H and Figure 2J to show that loss of EWS::FLI1 significantly perturbs p300 binding but not MLL4 at EWS::FLI1 enhancer loci.

Figure 3:

- Labeling could be improved in numerous places. For example, in figure 3G what are the tornado plots for? In the text it says histone acetylation, and in the figure legend it says H3K18ac- but it would help to have this labeled on the figure.

We have now added a label to this figure indicating it is H3K18ac ChIP-seq

- Also in this figure, specific enhancers were referred to as dHalo sensitive, which presumably refers to EWS::FLI1 sensitive. It may be best to label that way- as it is a bit confusing when called dHalo sensitive.

We agree with the reviewer and have now changed this to EWS::FLI1 sensitive for clarity.

- The authors should better explain why they examine the H3K18ac mark instead of H3K27ac?

This is an important point and we thank the reviewer for raising this. We included H3K18ac (alongside H3K27ac in most experiments) because it is dynamically linked to p300/CBP activity at active enhancers. We have now clarified our rationale in the text for assessing additional histone modifications beyond H3K27ac.

“We first examined how histone acetylation is affected at EWS::FLI1 enhancers following p300/CBP perturbation. Traditionally, p300/CBP has been associated with H3K27ac and enhancer activity (Creyghton et al., 2010). However, recent studies have shown that additional acetylation marks, including H3K18ac and acetylated H2B residues, are dynamically linked to p300/CBP function and active enhancer function (Weinert et al., 2018, Narita et al., 2023). Therefore, we expanded our analysis to include H3K18ac and H2BK120ac at EWS::FLI1 enhancers”

- Also, since the authors wanted to compare MLL3/4 activity to CBP-P300, why did they switch to using A673 and SKNMC (rather than A673 and EWS-502)?

We initially decided to focus on SKNMC cells later in the manuscript due to this cell line being one of the most widely used in the field and has been extensively studied in the context of EWS::FLI1 enhancers.

Despite this, we agree with the reviewer that it would be good to continue the assessment of EWS-502 cells throughout the manuscript, alongside A673 and SKNMC cells. We have now performed key experiments in the EWS-502 cells to demonstrate that p300/CBP function is critical in this context:

1. **Histone Modification Analysis:** We performed ChIP-seq and qPCR in EWS-502 cells following a 6-hour treatment with A485 and dCBP1, showing a reduction in both H3K27ac and H3K18ac at EWS::FLI1 enhancers (Figure EV2H-J).
2. **Transcriptomic Analysis:** RNA-seq after a 6-hour treatment with A485 and dCBP1 revealed that EWS::FLI1-regulated genes are highly enriched among the downregulated genes (Figure EV5C).
3. **Drug Synergy Experiments:** We assessed the combined effects of A485 and CCS1477, demonstrating that EWS-502 cells are highly sensitive to p300/CBP perturbation (Figure EV6B).

These findings reinforce the importance of p300/CBP function in EWS-502 cells, aligning with our observations in A673 and SKNMC.

Figure 4:

- To strengthen the claim about the scaffolding role of CBP/P300 they get from using CCS1477, it would be helpful to see KD/ genetic addback experiments, with specific domains of the protein deleted.

We appreciate that this is a key question in the field to discern differences between the catalytic and non-catalytic function of p300 and CBP. While this is an important aspect, our data demonstrate that catalytic activity is also essential for Ewing sarcoma cell growth (Figure 6A). Given this, we believe that reintroducing individual domains of p300/CBP would be insufficient to rescue cell viability. Additionally, our previous attempts to knockdown both p300 and CBP resulted in rapid cell death.

To address this comment we have clarified in the manuscript that p300 and CBP likely contribute to EWS::FLI1 gene target regulation through both catalytic and non-catalytic functions in both the results and discussion sections.

“While context-dependent functional differences between MLL3 and MLL4, as well as between p300 and CBP, have been highlighted and may exist in the Ewing sarcoma context, we will refer to these proteins collectively as MLL3/4 and p300/CBP throughout the paper, given the focus of dual targeting these proteins in our study”

“Importantly, p300 and CBP are distinct homologous proteins, and context-dependent functional differences have been observed in other cancer settings (Durbin et al., 2022, Martire et al., 2020). Our study focuses on the dual role of p300/CBP using tools that simultaneously target both proteins. We demonstrate that inhibiting both disrupts EWS::FLI1 target gene regulation and impairs Ewing sarcoma cell growth. Future studies aimed at distinguishing the specific and separate function of p300 and CBP in this context would provide valuable insights”

• **CBP and P300 are referred to as largely interchangeable. There are differences between the two proteins and the authors should explain why they focusing mostly on P300 and not CBP.**

We agree with the reviewer that CBP and p300 are distinct proteins with potential context-dependent functional differences. However, our study examines both p300 and CBP throughout. Specifically, we show that both are bound at most EWS::FLI1 enhancers (Figure 1D) and that their presence at the *NKX2-2* enhancer is disrupted following EWS::FLI1 degradation (Figure 2G-H, Figure EV2L). Additionally, we employ tools that target both p300 and CBP activity (A485 and dCBP1) and demonstrate that dCBP1 similarly reduces p300 and CBP protein levels (Figure 3A).

To address this point, we have clarified throughout the manuscript that p300 and CBP are distinct proteins. We have also added a note in the introduction acknowledging their potential context-specific differences and explaining why we refer to them collectively as p300/CBP for improved readability. Furthermore, we have expanded the discussion to highlight the importance of further investigating their distinct functional roles in this context in the future.

“While context-dependent functional differences between MLL3 and MLL4, as well as between p300 and CBP, have been highlighted and may exist in the Ewing sarcoma context, we will refer to these proteins collectively as MLL3/4 and p300/CBP throughout the paper, given the dual targeting of these complexes in our study. “

Figure 6:

- In figure 6, it would be helpful to run the models to determine if A-485, CCS1477, and combo treatment are additive, synergistic, or antagonistic.

We conducted drug titration experiments and analyzed the data using SynergyFinder 3.0 software. The synergy plot for all three cell lines (A673, SKNMC, and EWS-502) is shown in Figure EV6B. In all three cell lines, we observed a likely additive effect from combining A485 and CCS1477.

Minor issues to address:

1. In some parts of the paper, references are numbered rather than listed as author names etc.

We have corrected this throughout the paper.

Dear Dr. Armstrong,

Thank you for the submission of your revised manuscript. We have now received the enclosed reports from the referees that were asked to assess it, and I am happy to say that both support its publication now. Please address the last minor comment by referee 3. Also a few editorial requests will need to be addressed before we can proceed with the official acceptance of your manuscript.

- Please add up to 5 keywords to the ms file.
- Please place the Disclosure and Competing Interest Statement after the Acknowledgments
- The corresponding author needs to be marked on the title page, and also his email address needs to be listed there.
- The REFERENCE format needs to be corrected to the EMBO reports style: et al needs to be used after 10 author names.
- A callout for Fig. 6G is missing, please add.
- There are 12 EV tables, which need to be uploaded as individual files. I would suggest that you keep the following tables as EV tables: Table EV1, Table EV2, Table EV12; each table needs to have their own title. Tables EV3-EV9 need to be uploaded as individual Dataset files called Dataset EV1, etc. and also need to have their own title. Tables EV10 and EV11 can be part of the Reagents and Tools table. The single Appendix Figure can also be another EV figure so that the Appendix can be deleted. Please also update all ms callouts accordingly.
- The Source Data need to be uploaded separately, as one file per figure.
- Please remove the Related Ms files from the original submission.
- Please note that the exact p values are not provided in the legends of figures 2B, F, G, I; 3C, D, E, F, I, M, N, O; 4A, B, C, G, H, K; 5A, 6C, F; EV2 N; EV3 O, EV4 C. Please provide exact values as reasonable.
- Please indicate the statistical test used for data analysis in the legends of figures 5B, C, D, H; EV5 C.

EMBO press papers are accompanied online by A) a short (1-2 sentences) summary of the findings and their significance, B) 2-3 bullet points highlighting key results and C) a synopsis image that is exactly 550 pixels wide and 200-600 pixels high (the height is variable). The synopsis image should provide a sketch of the major findings, like a graphical abstract. Please note that text needs to be readable at the final size. Please send us this information along with the final manuscript.

I look forward to seeing a new revised version of your manuscript as soon as possible. Please use this link to submit your revision: <https://embor.msubmit.net/cgi-bin/main.plex>

Referee #1:

The authors have very carefully and adequately addressed all reviewers' points by adding additional data and clarifications to the manuscript.

Referee #3:

The authors have addressed all previous comments and the paper is significantly improved. It should be noted that EV6B is not labeled as such in the main manuscript (but could be found in the appendix).

The authors addressed the remaining editorial issues.

Dr. Scott Armstrong
Dana-Farber Cancer Institute
Department of Pediatric Oncology
United States

Dear Dr. Armstrong,

I am very pleased to accept your manuscript for publication in the next available issue of EMBO reports. Thank you for your contribution to our journal.

Yours sincerely,
